# The capsule and genetic background, rather than specific individual loci, strongly influence in vitro pneumococcal growth kinetics

Chrispin Chaguza[1,2,3,4]*, Daan W Arends[5], Stephanie W Lo[3], Indri Hapsari Putri[5], Anna York[1], John A Lees[6], Anne L Wyllie[1], Daniel M Weinberger[1], Stephen D Bentley[3], Marien I de Jonge[5†], Amelieke JH Cremers[5,7]*†

[1]Department of Epidemiology of Microbial Diseases, Yale School of Public Health, Yale University, New Haven, United States; [2]Yale Institute for Global Health, Yale University, New Haven, United States; [3]Parasites and Microbes Programme, Wellcome Sanger Institute, Wellcome Genome Campus, Hinxton, United Kingdom; [4]Department of Host-Microbe Interactions, St Jude Children's Research Hospital, Memphis, United States; [5]Laboratory of Medical Immunology, Radboud Institute of Molecular Life Sciences, Radboudumc Center for Infectious Diseases, Radboudumc, Nijmegen, Netherlands; [6]European Molecular Biology Laboratory–European Bioinformatics Institute, Cambridge, United Kingdom; [7]Department of Fundamental Microbiology, University of Lausanne, Lausanne, Switzerland

*For correspondence:
chrispin.chaguza@gmail.com (CC);
Amelieke.cremers@unil.ch (AJHC)

†These authors contributed equally to this work

Competing interest: The authors declare that no competing interests exist.

## eLife Assessment

This is an **important** study that examines the impact of *Streptococcus pneumoniae* genetics on its in vitro growth kinetics, aiming to identify potential targets for vaccines and therapeutics. The study identified significant variations in growth characteristics among capsular serotypes and lineages, linked to phylogeny and high heritability, but genome-wide association studies did not reveal specific genomic loci associated with growth features independent of the genetic background. The evidence supporting these findings is **convincing**.

**Abstract** Bacterial growth characteristics intrinsic to each strain can impact and influence gene expression, antibiotic susceptibility, and disease pathogenesis. However, little is known about specific genomic variations that influence these bacterial growth features. Here, we investigate the impact of *Streptococcus pneumoniae* genetics on its in vitro growth features to shed light on genes that may be important targets in the development of vaccines and therapeutics. We use statistical models to estimate growth features and demonstrate that they varied significantly across capsular serotypes and lineages, were strongly correlated with phylogeny, and showed high heritability, highlighting a strong genetic basis. Despite this, genome-wide association studies revealed no specific genomic loci statistically associated with the growth features independently of the genetic background, including those in the locus responsible for capsular polysaccharide synthesis. Our findings suggest that the serotype and lineage, as well as a combination of genomic loci, influence intrinsic pneumococcal growth kinetics, which may have implications for pneumococcal disease pathogenesis.

## Introduction

Once termed 'the captain of the men of death' by Sir William Osler in the 20th century, *Streptococcus pneumoniae*, or the pneumococcus, remains a significant cause of life-threatening invasive diseases, such as sepsis, pneumonia, and meningitis (*Ikuta et al., 2022*), despite the widespread implementation of life-saving serotype-specific polysaccharide pneumococcal conjugate vaccines (PCVs) (*von Gottberg et al., 2014*). The high capsular diversity, comprising ~100 serotypes with varying potential for colonising the nasopharynx and causing invasive diseases (*Ganaie et al., 2020*; *Brueggemann et al., 2003*) and mortality (*Grabenstein and Musey, 2014*; *Müller et al., 2022*; *Cohen et al., 2015*), demonstrates the remarkable genetic diversity of the pneumococcus. Such high genetic diversity reflects the plasticity of the pneumococcal genome, which readily exchanges genetic material with other strains through recombination and horizontal gene transfer (*Donati et al., 2010*). Capsule switching, resulting from the swapping of the capsule biosynthesis (*cps*) loci, which encode the extracellular polysaccharide capsule that determines the capsular serotype, exemplifies genetic exchange between pneumococcal strains (*Mostowy et al., 2017*). While serotypes typically exist in specific lineages (*Croucher et al., 2013*), such as those defined by the global pneumococcal sequence cluster (GPSC) nomenclature (*Gladstone et al., 2019*), some serotypes are expressed across distinct lineages, while some lineages express multiple serotypes, reflecting capsule switching (*Croucher et al., 2015a*). However, the impact of individual capsular serotypes, genetic background, and their combinations on the bacterial phenotypes, especially those associated with disease pathogenesis and clinical manifestations, remains less understood.

The pneumococcus possesses an arsenal of virulence factors. The extracellular polysaccharide capsule is a primary virulence factor crucial for pneumococcal transmissibility, survival, and pathogenicity in the human host (*Kadioglu et al., 2008*; *Hathaway et al., 2016*; *Hamaguchi et al., 2018*). Recent studies on the pneumococcus and other bacterial pathogens suggest that variability in growth kinetics affects disease pathogenesis, clinical manifestation, and antibiotic therapy. For example, differential pneumococcal growth features appear to correlate with fitness for colonisation and invasive disease (*Im et al., 2022*), clinical manifestation (*Im et al., 2022*; *Arends et al., 2022*; *Hathaway et al., 2012*), and response to environmental conditions (*Hathaway et al., 2012*; *Tóthpál et al., 2019*; *Small et al., 1986*). Studies of other bacterial pathogens suggest that growth rate influences gene and protein expression (*Klumpp et al., 2009*; *Nilsson et al., 1984*), adhesion (*Rogers et al., 1984*), bacterial competition (*Russell et al., 1979*), plasmid replication (*Lin-Chao and Bremer, 1986*), bacteriophage dynamics (*Nabergoj et al., 2018*), and antibiotic accumulation and efficacy (*Łapińska et al., 2022*; *Smirnova and Oktyabrsky, 2018*; *Tuomanen et al., 1986*). Similarly, experimental and mathematical modelling studies have demonstrated that bacterial growth correlates with bacterial load and host-mediated cell lysis, impacting dispersal to different host cells and tissues during systemic infection (*Grant et al., 2009*). Previous studies of pneumococcus and other bacteria have suggested the influence of environmental factors (*Tóthpál et al., 2019*; *Small et al., 1986*) and bacterial-specific factors, including capsular serotype (*Hathaway et al., 2012*; *Müller et al., 2020*) and the presence of plasmids (*Gulig and Doyle, 1993*), on bacterial growth phenotypes. These growth phenotypes may consequently impact the pathogenesis of invasive pneumococcal disease (*Im et al., 2022*; *Arends et al., 2022*; *Hathaway et al., 2012*). However, the specific pneumococcal genomic variations affecting growth kinetics remain less understood.

Studying bacterial growth kinetics sheds light on the influence of genes, genomic variation, and growth media conditions on bacterial fitness (*Shlla et al., 2021*; *Zhang et al., 2021*). However, studying bacterial growth kinetics in vivo remains intrinsically challenging due to the complexity of the conditions in the hosts, although innovative approaches are being developed (*Gao and Li, 2018*; *Korem et al., 2015*). On the other hand, in vitro methods provide a valuable alternative for assessing the impact of specific bacterial and environmental factors on bacterial fitness. Although these in vitro methods have yielded remarkable insights into bacterial fitness, they have primarily provided strain-specific rather than population-level insights due to the analysis of a small number of isolates from a limited number of genetic backgrounds.

Here, we studied the impact of pneumococcal genetics on in vitro growth kinetics to understand the influence of bacterial genetic variation on pneumococcal pathogenesis. Specifically, we performed high-throughput quantification of the in vitro growth kinetics and whole-genome sequencing of 348 invasive pneumococcal isolates from the Netherlands to determine the impact of genomic variation

on growth kinetics. We hypothesised that the capsular serotype, genetic background, and specific individual genomic loci independent of the genetic background influence in vitro pneumococcal growth kinetics features or parameters derived from the growth curves. We tested this hypothesis through a series of statistical and comparative genomics analyses, including a bacterial genome-wide association study (GWAS) approach. Our study provides insights into the influence of capsular serotype, genetic background or lineage, and individual genomic loci on pneumococcal growth kinetics – a fundamental trait that potentially influences nasopharyngeal colonisation, invasive disease pathogenesis, and clinical manifestations.

## Results

### Population structure of a diverse collection of Dutch invasive pneumococcal isolates and derivation of their growth kinetics

We aimed to understand the impact of the capsular serotype, genetic background, and specific genomic loci, independent of the clonal population structure, on the pneumococcal growth features (*Figure 1*, *Supplementary file 1*). Therefore, we investigated the in vitro growth kinetics of 348 pneumococcal isolates collected from patients with invasive pneumococcal diseases via the Pneumococcal Bacteraemia Collection Nijmegen (PBCN) cohort study at two hospitals in Nijmegen, the Netherlands, before (2000–2006) and after (2007–2011) the introduction of the seven-valent PCV [PCV7] into the Dutch paediatric immunisation programme (*Cremers et al., 2014*). Our isolates represent 34 capsular serotypes, 40 pneumococcal lineages, as defined by the GPSC nomenclature (*Gladstone et al., 2019*), and 99 sequence types (STs) specified by the pneumococcal multilocus sequence typing (MLST) scheme (*Enright and Spratt, 1998*). An interactive and annotated maximum likelihood phylogeny of the isolates used in this study is available online at the Microreact platform for open data visualisation and sharing for genomic epidemiology (https://microreact.org/project/wxmdblfgprepwugvbbtaq7). We estimated a Simpson diversity index of 0.932 (95% confidence interval [CI]: 0.924–0.940), 0.937 (95% CI: 0.928–0.945), and 0.958 (95% CI: 0.949–0.966) for the serotypes, lineages, and STs, respectively. The high richness and diversity of pneumococcal serotypes and lineages highlighted a substantial population-level diversity of the study isolates in the Netherlands, consistent with earlier studies of the same population (*Cremers et al., 2019*; *Cremers et al., 2015*).

### Capsular serotype and genetic background significantly impact pneumococcal growth kinetics

To explore the impact of capsular serotype and genetic background or lineage on the pneumococcal growth features estimated from a fitted growth curve, namely lag-phase duration, maximum growth density ($H_{max}$), maximum growth change ($\Delta H$), and average growth rate ($r$) (*Figures 1 and 2*, *Figure 2— figure supplements 1–19*, and *Supplementary file 1*), we assessed the relationship between these growth features and serotypes and lineages associated with the study isolates (*Figure 3a*). In general, we found significant variability of the growth features across serotypes: average growth rate, maximum growth density, maximum growth change, and lag phase duration ($P<0.0001$; Kruskal–Wallis test) consistent with previous studies (*Hathaway et al., 2012*; *Tóthpál et al., 2019*; *Figure 3*). We observed similar patterns across different pneumococcal lineages (*Figure 3f–i* and *Supplementary file 2*).

Recombination and horizontal gene transfer may lead to the replacement of part or the entire capsule biosynthesis locus, resulting in the production of a polysaccharide capsule distinct from the original one, a phenomenon called capsule-switching (*Croucher et al., 2011*). Consequently, some lineages are associated with multiple serotypes, and similarly, a single serotype can be expressed in distinct genetic backgrounds (*Croucher et al., 2015a*). Examples of lineages containing multiple serotypes included GPSC3, GPSC4, and GPSC7, and likewise, serotypes, including 3, 19A, and 19F, were expressed in multiple genetic backgrounds, which also express other serotypes (*Figure 3a*). Overall, the average growth rate for serotypes expressed in different genetic backgrounds broadly showed similar patterns. For example, lineages expressing serotype 14 had consistently higher average growth rates. In contrast, lineages expressing serotype 3 showed lower rates, consistent with the notion that capsule production interferes with pneumococcal growth (*Hathaway et al., 2012*; *Figure 3b*). Therefore, we investigated the impact of the capsular serotype and genetic background on the estimated pneumococcal growth features.

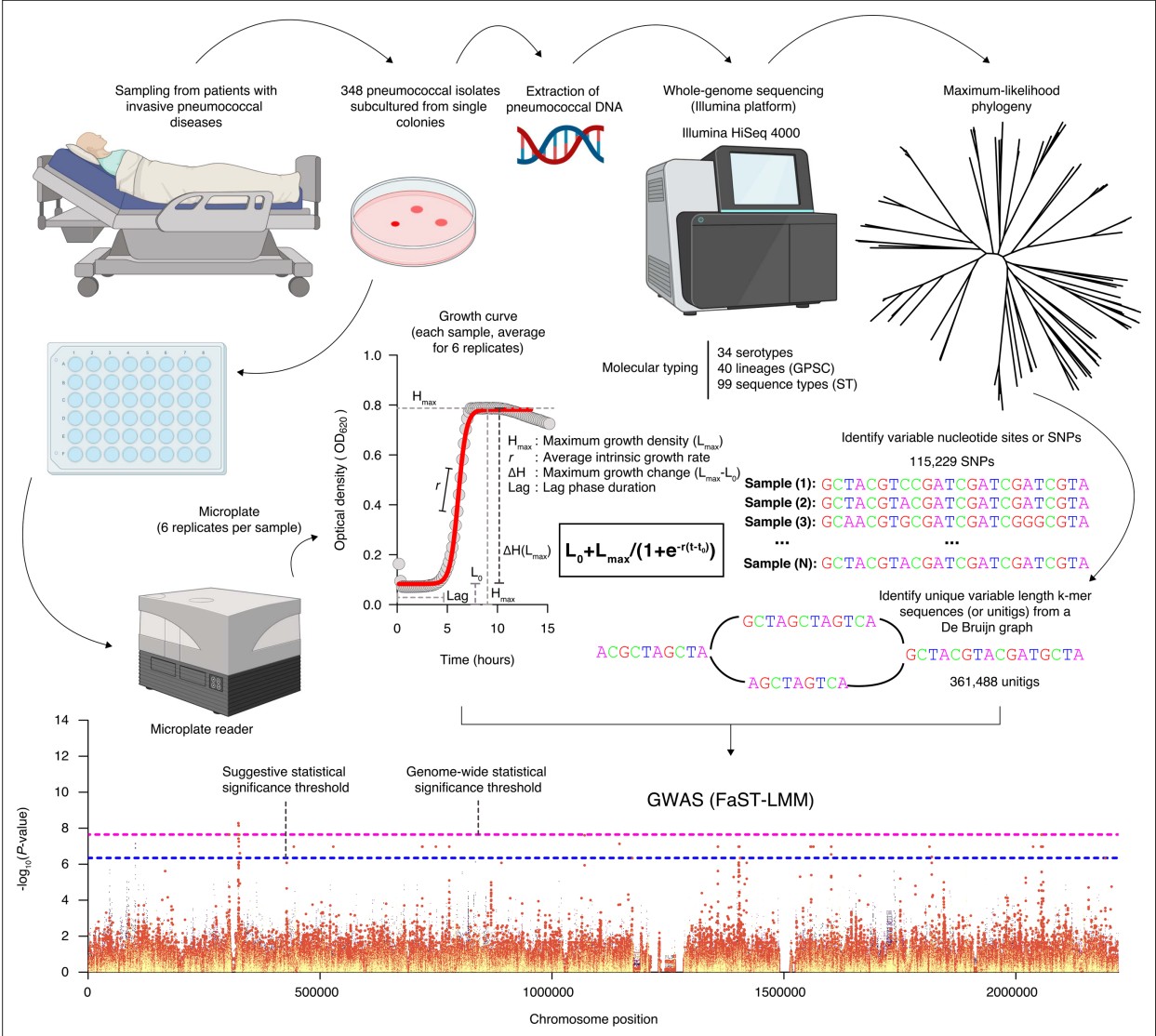

**Figure 1.** Schematic overview of high-throughput quantification of in vitro pneumococcal growth kinetics and the genome-wide discovery of individual genomic variation influencing in vitro pneumococcal growth. Population genomic approaches using genome-wide association study (GWAS) offer a robust way to identify genomic variation influencing the growth dynamics of the pneumococcus. First, we collected 348 pneumococcal blood culture isolates that were obtained from 348 patients admitted to the hospital with invasive pneumococcal disease. The isolates were inoculated in a liquid medium in a highly standardised manner followed by measurement of the growth kinetics after every 10 minutes for slightly over 15 hours using a microplate reader (**Arends et al., 2022**). We then fitted a logistic function (see 'Materials and methods') to measure growth features, namely, the average growth rate ($r$), maximum growth density ($H_{max}$) defined as $L_{max}$, lag phase duration, and maximum growth change ($\Delta H$), defined as the difference between the maximum and minimum density ($L_{max}-L_{min}$). Additionally, genomic DNA was extracted from the cultured isolates for whole-genome sequencing using the Illumina HiSeq 4000 sequencing platform. The generated sequencing data was used to reconstruct a maximum-likelihood phylogenetic tree of the isolates for downstream analyses, for example, the assessment of phylogenetic signals for the estimated growth features. We also performed de novo assembly of the sequencing reads to generate contigs, which were used to map against the ATCC 700669 pneumococcal reference genome (GenBank accession: NC_011900) to identify consensus single nucleotide polymorphisms (SNPs). We also generated presence, and absence patterns for unitigs or variable length $k$-mer sequences based on the draft assembled contigs. The SNPs were used to generate the phylogeny of the isolates while the unitigs were used in a GWAS to identify variants associated with the estimated pneumococcal growth features. Created with BioRender.com.

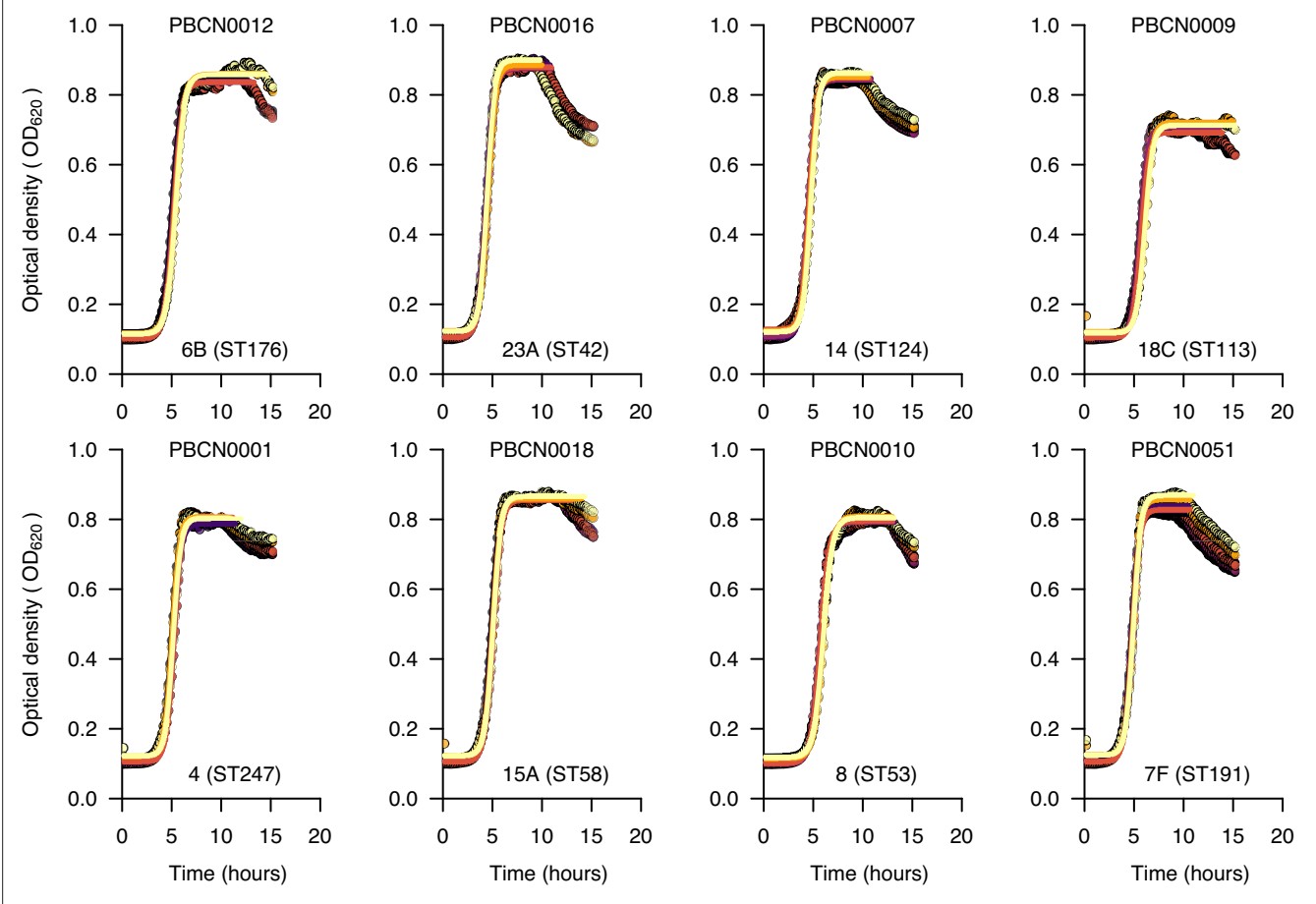

**Figure 2.** Growth curves for selected serotypes show the in vitro pneumococcal growth dynamics and estimated growth features inferred by mathematical modelling. The growth curves above show the optical density at 620 nm monitored over 15 hours for an example dataset of 20 pneumococcal isolates representing different serotypes and sequence types (STs) based on the pneumococcal multilocus sequence typing (MLST) scheme. We used the nonlinear optimisation algorithm to identify a combination of parameters with the best fit to a logistic curve based on the lowest residual sum of squares for the predicted value based on the model relative to the observed growth data (see 'Materials and methods'). We fitted separate curves based on a subset of points for each replicate up to the end of the stationary phase or end of the experiment, whichever came first. We did this to get a better fit for the logistic curve during the growth phase, as it does not allow for negative growth, which happened after the stationary phase. We fitted the curves to infer the growth features for each of the six replicates. The averaged estimates for each replicate were then used for the downstream analyses. Additional plots showing fitted curves for all the isolates are shown in *Figure 2—figure supplements 1–19*.

The online version of this article includes the following figure supplement(s) for figure 2:

**Figure supplement 1.** Growth curves for selected serotypes show the in vitro pneumococcal growth dynamics and estimated growth features inferred by fitting a statistical model.

**Figure supplement 2.** Growth curves for selected serotypes show the in vitro pneumococcal growth dynamics and estimated growth features inferred by fitting a statistical model.

**Figure supplement 3.** Growth curves for selected serotypes show the in vitro pneumococcal growth dynamics and estimated growth features inferred by fitting a statistical model.

**Figure supplement 4.** Growth curves for selected serotypes show the in vitro pneumococcal growth dynamics and estimated growth features inferred by fitting a statistical model.

**Figure supplement 5.** Growth curves for selected serotypes show the in vitro pneumococcal growth dynamics and estimated growth features inferred by fitting a statistical model.

**Figure supplement 6.** Growth curves for selected serotypes show the in vitro pneumococcal growth dynamics and estimated growth features inferred by fitting a statistical model.

**Figure supplement 7.** Growth curves for selected serotypes show the in vitro pneumococcal growth dynamics and estimated growth features inferred by fitting a statistical model.

*Figure 2 continued on next page*

*Figure 2 continued*

**Figure supplement 8.** Growth curves for selected serotypes show the in vitro pneumococcal growth dynamics and estimated growth features inferred by fitting a statistical model.

**Figure supplement 9.** Growth curves for selected serotypes show the in vitro pneumococcal growth dynamics and estimated growth features inferred by fitting a statistical model.

**Figure supplement 10.** Growth curves for selected serotypes show the in vitro pneumococcal growth dynamics and estimated growth features inferred by fitting a statistical model.

**Figure supplement 11.** Growth curves for selected serotypes show the in vitro pneumococcal growth dynamics and estimated growth features inferred by fitting a statistical model.

**Figure supplement 12.** Growth curves for selected serotypes show the in vitro pneumococcal growth dynamics and estimated growth features inferred by fitting a statistical model.

**Figure supplement 13.** Growth curves for selected serotypes show the in vitro pneumococcal growth dynamics and estimated growth features inferred by fitting a statistical model.

**Figure supplement 14.** Growth curves for selected serotypes show the in vitro pneumococcal growth dynamics and estimated growth features inferred by fitting a statistical model.

**Figure supplement 15.** Growth curves for selected serotypes show the in vitro pneumococcal growth dynamics and estimated growth features inferred by fitting a statistical model.

**Figure supplement 16.** Growth curves for selected serotypes show the in vitro pneumococcal growth dynamics and estimated growth features inferred by fitting a statistical model.

**Figure supplement 17.** Growth curves for selected serotypes show the in vitro pneumococcal growth dynamics and estimated growth features inferred by fitting a statistical model.

**Figure supplement 18.** Growth curves for selected serotypes show the in vitro pneumococcal growth dynamics and estimated growth features inferred by fitting a statistical model.

**Figure supplement 19.** Growth curves for selected serotypes show the in vitro pneumococcal growth dynamics and estimated growth features inferred by fitting a statistical model.

First, we identified lineages that expressed multiple serotypes and serotypes found across multiple lineages. To ensure a robust statistical comparison, we only focused on serotype-lineage combinations with at least three sequenced isolates in our study. We identified two lineages, GPSC3 and GPSC7, with sufficient sequenced isolates for comparison, which expressed multiple serotypes (*Figure 3j*). We observed higher growth rates for serotype 33F than serotype 8 (=0.0007; Kruskal–Wallis test).

In contrast, we found no statistically significant differences in growth rates between serotypes 23A and 23F in lineage GPSC7 ($P=0.2568$; Kruskal–Wallis test). We speculated that the absence of statistically significant differences between serotypes 23A and 23F reflected their presence in the same serogroup; therefore, they had similar capsule properties. Interestingly, we observed similar patterns for the maximum growth density, maximum growth change, and lag phase duration features between serotypes on the GPSC3 and GPSC7 lineages (*Figure 3k–m* and *Supplementary file 2*). Next, we investigated the growth rates of the same serotypes expressed in distinct genetic backgrounds. Although some serotypes exhibited similar growth rates across distinct genetic backgrounds, such as serotype 3 ($P=0.4599$; Kruskal–Wallis test), others showed different growth rates across different lineages, as seen with serotypes 14 ($P=0.0006$; Kruskal–Wallis test) and 19F ($P=0.0105$; Kruskal–Wallis test). We noted similar patterns for the other growth features: maximum growth density, maximum growth change, and lag phase duration (*Figure 3n–q*).

These findings demonstrate that the capsular serotype and genetic background significantly impact pneumococcal growth kinetics. The resultant growth features likely depend on the interaction between these factors, whereby some combinations may result in more pronounced growth effects than others.

## Pneumococcal growth kinetics show heritability and correlation with the phylogeny of the isolates

Based on the findings that both capsular serotype and genetic background are associated with growth kinetics, we hypothesised that the estimated pneumococcal growth features exhibit a high correlation between the pneumococcal growth features and the phylogeny, that is, phylogenetic signal. To assess this, we first performed ancestral state reconstruction to map the derived pneumococcal

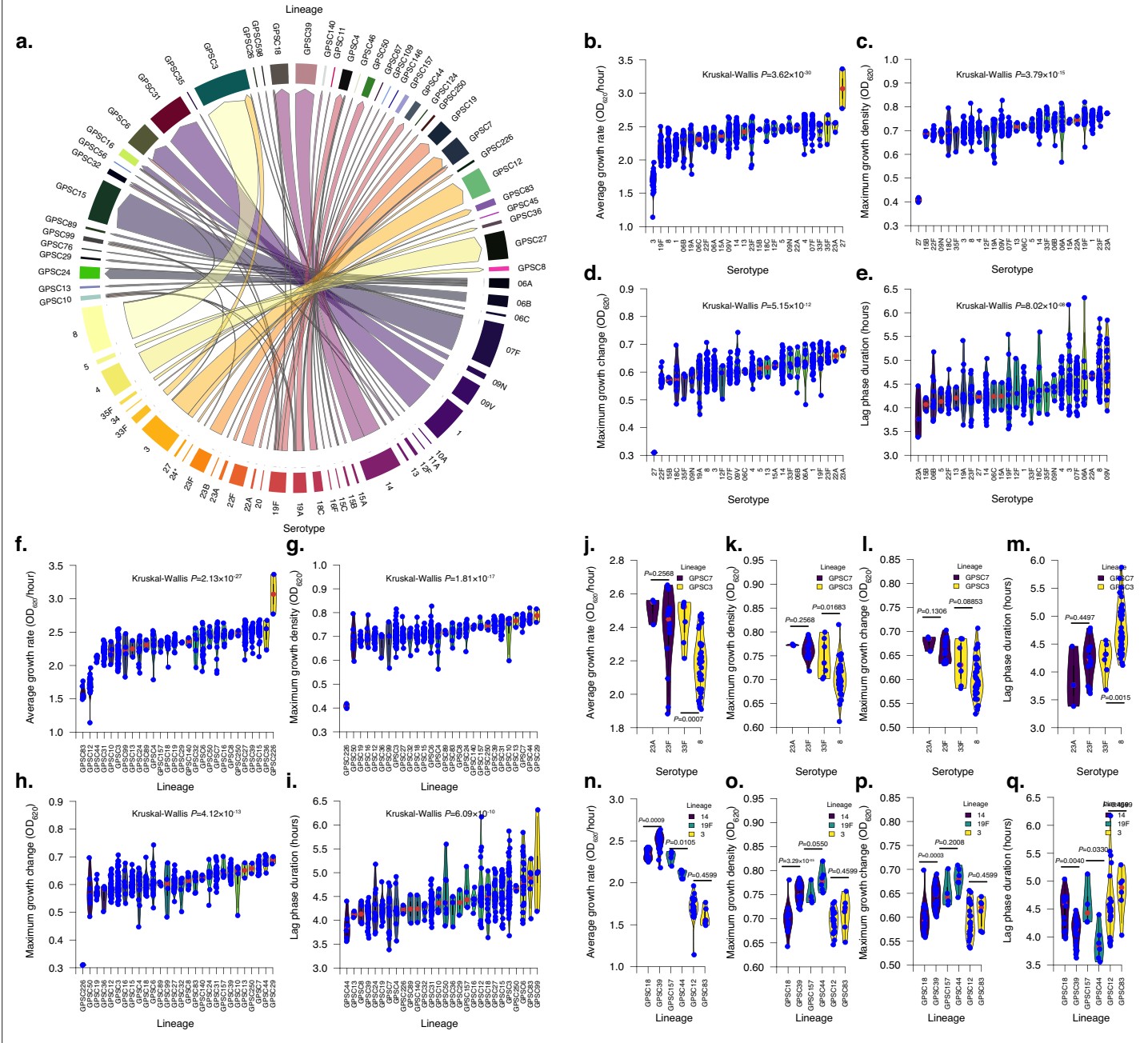

**Figure 3.** The in vitro pneumococcal growth features show variable patterns dependent on the capsular serotype and genetic background or lineage. (**a**) Chord diagram showing the association between pneumococcal capsular serotypes and lineages for the Dutch isolates included in this study. The serotypes were determined using in silico approaches based on the sequencing data using SeroBA (*Epping et al., 2018*), while the lineages were defined using the PopPUNK framework (*Lees et al., 2019b*) based on the global pneumococcal sequencing cluster (GPSC) lineage nomenclature (*Gladstone et al., 2019*). The thickness of the connecting lines between serotypes and lineages in the chord diagram represents the percentage of isolates belonging to each serotype and lineage. Due to capsule switching, some serotypes were associated with multiple lineages, and similarly, some lineages were associated with multiple serotypes. The violin plot shows the distribution of the estimated average growth rate across different serotypes. The distribution of the estimated growth features varied across serotypes with at least three isolates, as shown by the violin plots for the (**b**) average growth rate, (**c**) maximum growth change, (**d**) maximum growth change across serotypes, (**e**) lag phase duration growth features. Similarly, the distribution of the estimated growth features varied across lineages with at least three isolates, as shown by the violin plots for the (**f**) average growth rate, (**g**) maximum growth change, (**h**) maximum growth density, and (**i**) lag phase duration growth features. A comparison of the growth features between two serotypes belonging to the same lineages with at least three isolates is shown for the (**j**) growth rate, (**k**) maximum growth, (**l**) maximum growth change, and (**m**) the lag phase duration features. A comparison of the growth features between two lineages expressing the same serotype containing at least three isolates is shown for the (**n**) growth rates, (**o**) maximum growth, (**p**) maximum growth change, and (**q**) the lag phase features. Additional information regarding the isolates and their estimated growth features or parameters is available in *Supplementary file 2*.

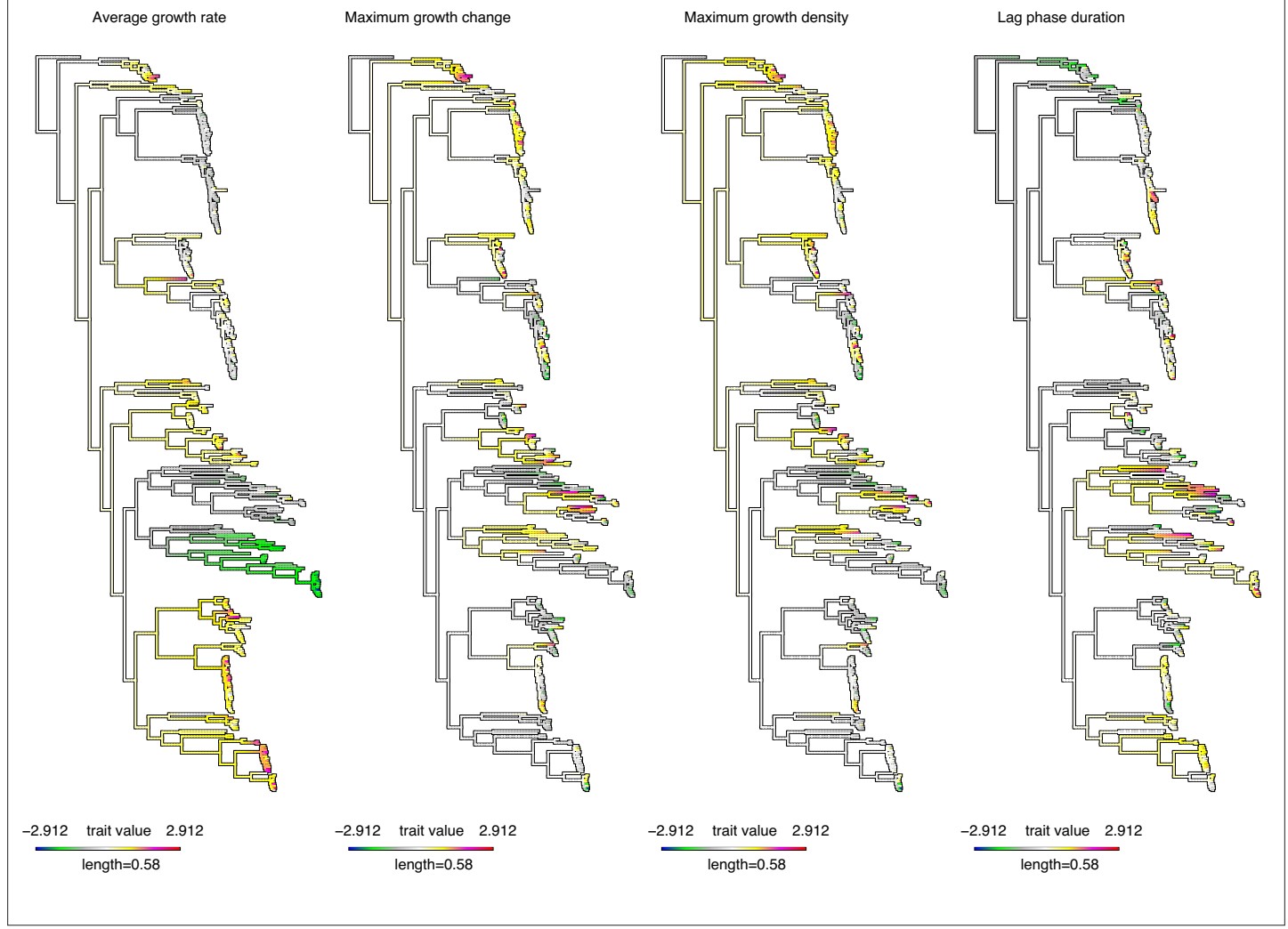

**Figure 4.** Ancestral reconstruction reveals a strong correlation between the estimated in vitro pneumococcal growth features and the whole-genome phylogeny of the isolates. The evolution of the normalised estimated continuous growth features, namely average growth rate, maximum growth change, maximum growth density, and lag phase duration, was assessed by estimating the states of the internal nodes of the maximum likelihood phylogenetic tree of the isolates using a maximum likelihood ancestral reconstruction approach (*Revell, 2012*). The length of the scale bar shows the number of nucleotide substitutions per site, while the colours represent the lowest and highest unnormalised growth features. The length of the scale bar shows the number of nucleotide substitutions per site, while the colours represent the lowest and highest unnormalised growth features. A phylogenetic tree showing bootstrap support values on the internal nodes of the tree is provided in *Supplementary file 3*.

The online version of this article includes the following figure supplement(s) for figure 4:

**Figure supplement 1.** Correlation between the normalised and unnormalised growth parameters.

**Figure supplement 2.** Ancestral reconstruction reveals a strong correlation between the estimated in vitro pneumococcal growth features and the whole-genome phylogeny of the isolates.

growth features or parameters across the whole genome phylogeny of the isolates. By reconstructing the estimated growth features on the internal nodes of the phylogeny, we found high but variable associations between the growth features and different genetic backgrounds (*Figure 4*, *Figure 4— figure supplements 1 and 2*, and *Supplementary file 2*). Broadly, genetically similar isolates from the same phylogenetic lineages or genetic backgrounds exhibited similar growth kinetics, varying in degree depending on the specific growth feature. Altogether, these findings suggest a strong phylo-genetic signal in the growth features.

Next, we calculated the phylogenetic signal of each pneumococcal growth feature using Pagel's $\lambda$ statistic (*Pagel, 1999*), a metric typically ranging from 0 to 1 that evaluates the correlation between given traits and the phylogeny, with higher values indicating a stronger phylogenetic signal. The

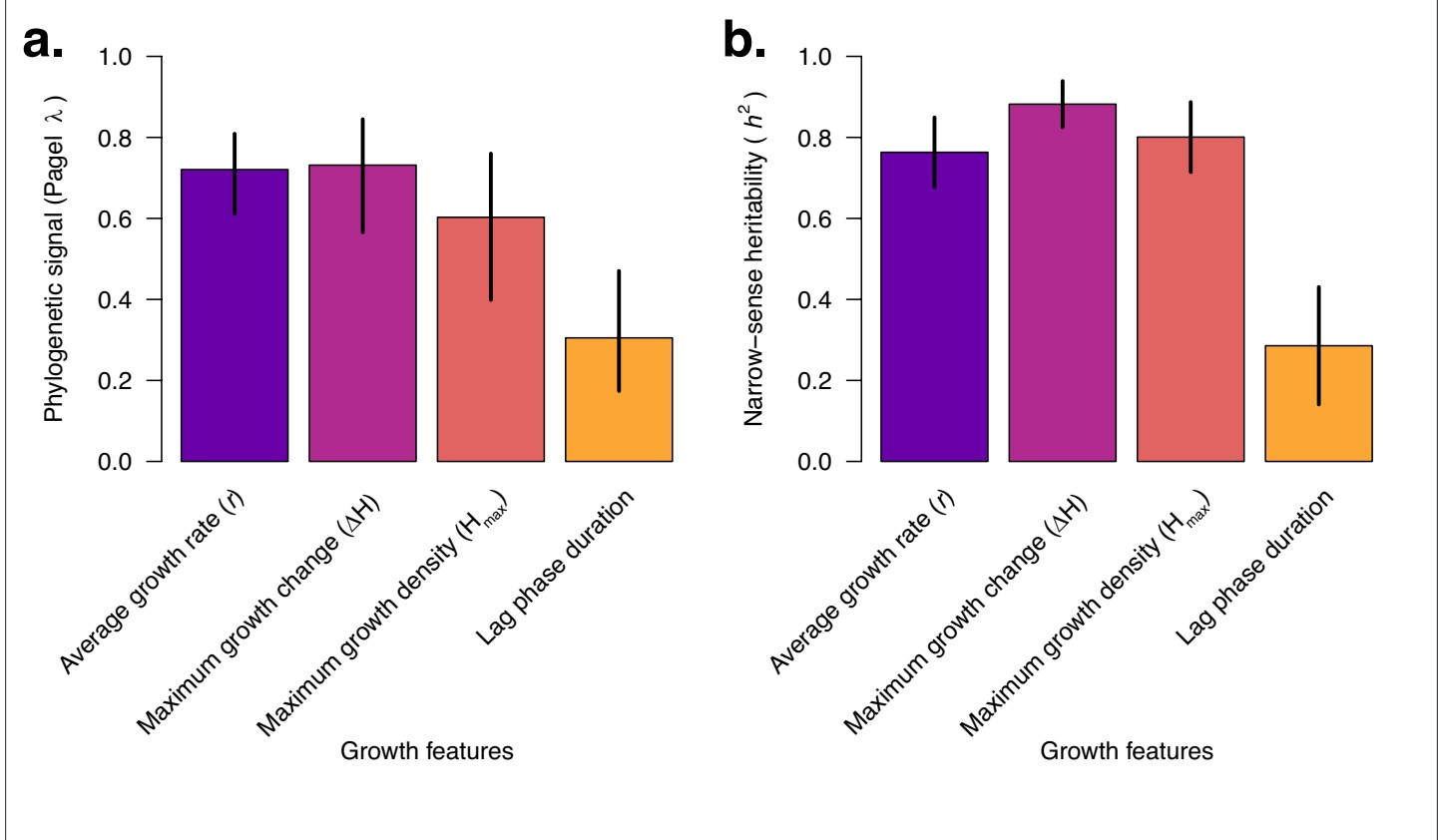

**Figure 5.** The estimated in vitro pneumococcal growth features show high phylogenetic signals and narrow-sense heritability. (**a**) Bar plot showing the phylogenetic signal of the growth features using Pagel's $\lambda$ statistic (**Pagel, 1999**). Pagel's $\lambda$ measures the correlated evolution of the traits with the phylogenetic tree of the isolates. Pagel's $\lambda$ estimates close to zero indicate no phylogenetic signal, i.e., the growth features were independent of the phylogeny, while values close to one show a strong phylogenetic signal. The average growth rate, maximum growth change, and maximum growth density showed the highest phylogenetic signal, while the lag phase duration feature showed an intermediate signal. In contrast, the average growth rate had the lowest phylogenetic signal, i.e., evolved more independently of the phylogeny. (**b**) Bar plot showing the narrow-sense heritability ($h^2$) of the pneumococcal growth features based on unitig sequences. The narrow-sense heritability corresponds to the amount of variability in the traits, i.e., growth features, explained by the pneumococcal genetic variation. The heritability is the proportion of phenotypic variation explained by the bacterial genetics (PVE) parameter using the linear mixed effects model implemented in GEMMA (**Zhou and Stephens, 2012**). A heritability value close to zero implies low heritability or minimal contribution of genetics to the variability of the traits. In contrast, values close to one suggest a strong influence of genetics on growth features. There was a substantial impact of pneumococcal genetics on all the growth features, with the highest influence seen for the average growth rate, maximum growth change, and maximum growth density, with a slightly lower contribution for the lag phase duration and the average growth rate . All the error bars represent the 95% confidence intervals. Additional information is provided in **Supplementary file 3**.

amount of the phylogenetic signals based on the normal transformed growth features was highest for the average growth rate (Pagel's $\lambda$ =0.72, 95% CI: 0.61–0.81; $P<2.08 \times 10^{-64}$), maximum growth change (Pagel's $\lambda$ =0.60, 95% CI: 0.40–0.76; $P<1.53 \times 10^{-16}$), and maximum growth density (Pagel's $\lambda$ =0.73, 95% CI: 0.57–0.84; $P<4.18 \times 10^{-26}$), while the lag phase duration showed a lower phylogenetic signal (Pagel's $\lambda$ =0.30, 95% CI: 0.17–0.47; $<5.42 \times 10^{-16}$) (**Figure 5a**). These findings suggest that there is a heterogeneous correlation of the growth features with the phylogeny.

To quantify the variability in pneumococcal growth kinetics explained by genotype, we calculated the narrow-sense heritability ($h^2$) for the four normal-transformed estimated growth features and the average growth rate . We found strong, though not equal, evidence for a genetic basis for all the estimated growth features: average growth rate ($h^2$=0.76, 95% CI: 0.68–0.85), maximum growth change ($h^2$=0.80, 95% CI: 0.71–0.89), average growth density ($h^2$=0.88, 95% CI: 0.83–0.94), and lag phase duration ($h^2$=0.29, 95% CI: 0.14–0.43) (**Figure 5b**, **Supplementary file 2**). To further understand the impact of the capsular serotype and genetic background, we fitted a linear mixed effects model for each growth parameter with serotype and lineage (defined by the GPSC nomenclature; **Gladstone et al., 2019**) as random effects to estimate the amount of variance of the growth features explained

by the serotype ($h^2_{serotype}$) and lineage or GPSC ($h^2_{GPSC}$) calculated as the ratio of the variance of the serotype and lineage components relative to the total variance, including the residual or environmental effects. We calculated $h^2_{serotype}=0.47$ and $h^2_{GPSC}=0.11$ for the average growth rate, $h^2_{serotype}=0.13$ and $h^2_{GPSC}=0.28$ for the maximum growth change, $h^2_{serotype}=0.29$ and $h^2_{GPSC}=0.46$ for the maximum growth, $h^2_{serotype}=0.08$ and $h^2_{GPSC}=0.39$ for the lag phase duration feature.

Altogether, these findings demonstrate a substantial impact of serotype and lineage on the variation in pneumococcal growth features, whereby most serotypes tend to have similar characteristics regardless of the genetic background. Similarly, some genetic backgrounds tend to exhibit similar phenotypic patterns regardless of the serotypes, but the serotype appears to have a more significant influence on the growth characteristics.

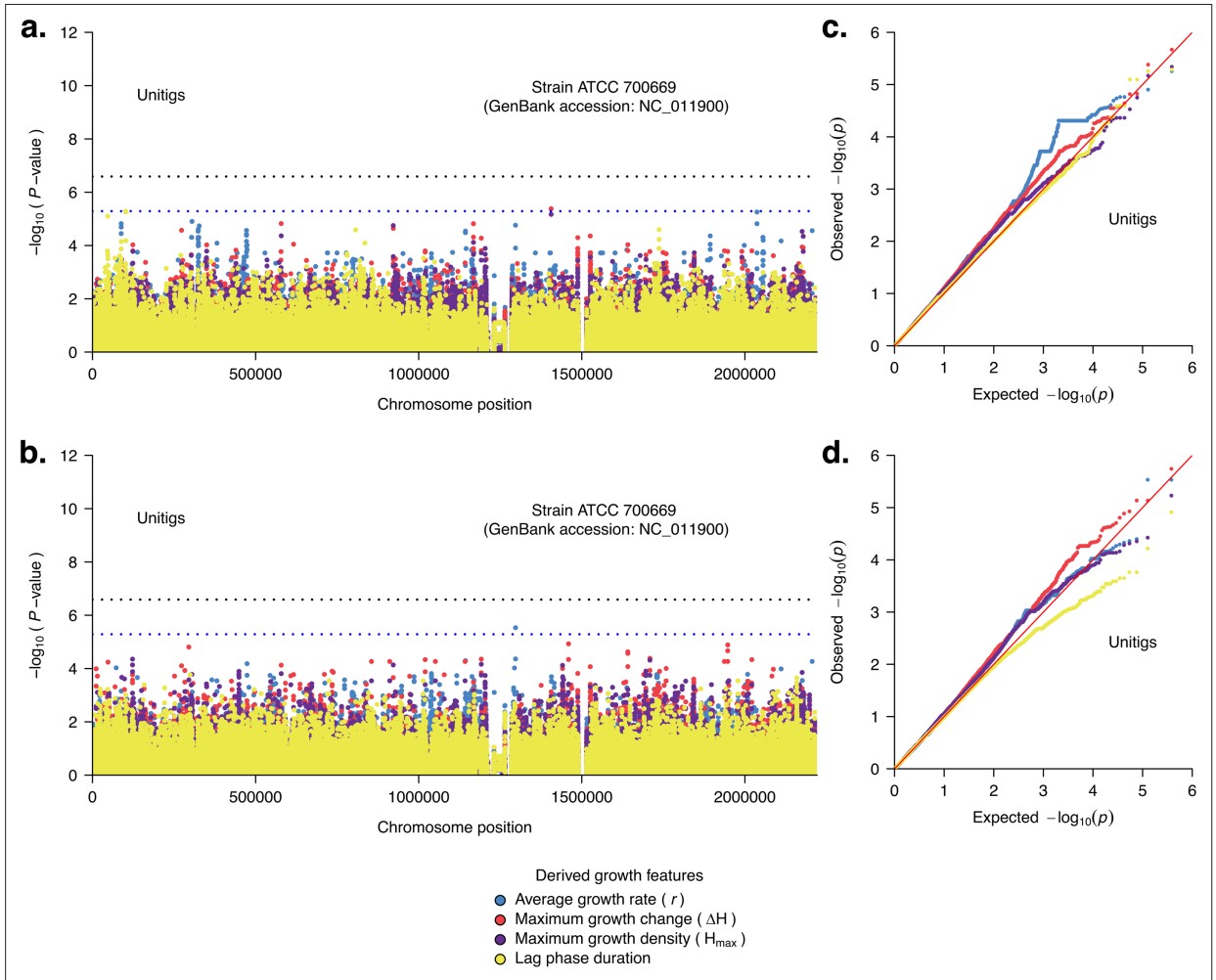

**Figure 6.** Pneumococcal genetic variation associated with the in vitro growth features or parameters of the isolates. Manhattan plot showing the statistical significance of the hits from the genome-wide association study (GWAS) analysis based on unitigs (**a**) without adjusting for the serotype and (**b**) after adjusting for the serotype. The statistical significance (*P*-value) was inferred based on the likelihood ratio test using Pyseer (***Lees et al., 2018a***) mapped against the ATCC 700669 pneumococcal reference genome (GenBank accession: NC_011900; ***Croucher et al., 2009***) to identify their location in the pneumococcal genome. The dots in the Manhattan plots with different colours represent GWAS for different growth features, as shown in the key. The red dotted line designates the genome-wide threshold for considering a variant as significantly associated, statistically, with the growth features. The blue dotted line represents the threshold for classifying variants as suggestively associated with the growth features. (**c**) Quantile–quantile plots showing the relationship between the observed statistical significance and the expected statistical significance for the GWAS of growth features not adjusting for the serotype. (**d**) Quantile–quantile (QQ) plots showing the relationship between the observed statistical significance and the expected statistical significance for the GWAS of growth features after adjusting for the capsular serotype.

## No individual genomic loci are associated with pneumococcal growth features

After finding moderate to high narrow-sense heritability of the growth features, we then undertook a GWAS analysis to identify specific genomic loci associated with these features, independent of the strains' genetic background using a linear mixed-effects model. We performed separate GWAS analyses of the maximum growth change, maximum growth density, maximum growth change, and lag phase duration phenotypes derived from the growth curve. We conducted the GWAS using variable-length *k*-mer sequences or unitigs to capture genetic variation in the pneumococcal genomes at different levels, including SNPs, multi-allelic sites, and insertions and deletions (*Jaillard et al., 2018*; *Lees et al., 2020*). We found no individual unitigs associated with the derived growth feature phenotypes (*Figure 6a*). Furthermore, repeating the GWAS analyses adjusting for the serotype revealed no statistically significant hits associated with the derived growth phenotypes, suggesting that the variants in the pneumococcal strains are correlated with the growth kinetics independent of the serotype (*Figure 6b*). Overall, the observed and expected statistical significance values in the QQ-plots for our study revealed no apparent issues with the population structure of the isolates, which is a significant confounder in similar studies (*Power et al., 2017*; *Figure 6c and d*). While the capsule is biologically relevant for pneumococcal growth (*Figure 3b–q*), the absence of individual loci even within the capsule biosynthesis locus with elevated signals associated with the growth features may be a mere consequence of adjusting for the population structure as pneumococcal serotypes are typically associated with specific lineages (*Figure 3a*). Overall, these findings suggest that the contribution of individual genomic loci in influencing pneumococcal growth kinetics does not stand out, potentially reflecting the polygenic nature of the phenotype and cumulative effect of the genetic background or lineage.

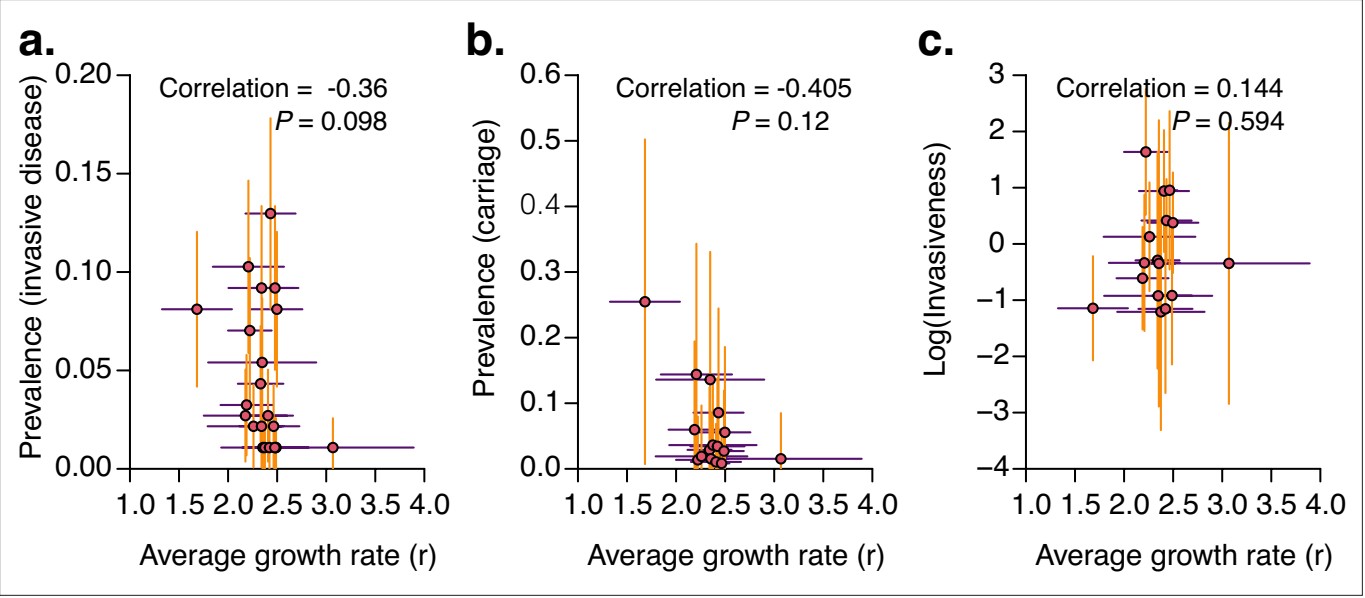

**Figure 7.** Association between average in vitro pneumococcal growth rate with the prevalence of serotypes in invasive disease, nasopharyngeal carriage, and invasiveness. (**a**) Scatter plot showing the relationship between the prevalence of serotypes among invasive disease isolates collected before the introduction of PCV7 in the Netherlands and the average in vitro growth rate (*r*) of the isolates inferred as shown in *Figure 1*. A Spearman correlation test showed no evidence of an association between the growth rate and prevalence of serotypes among the invasive disease isolates. (**b**) Scatter plot showing the relationship between the inferred prevalence of serotypes in the nasopharyngeal carriage before the introduction of PCV7 in the Netherlands and the inferred average in vitro growth rate. Due to the unavailability of the nasopharyngeal carriage data, we imputed the prevalence of the serotypes in carriage by dividing the prevalence in invasive disease by the average invasiveness of each serotype. The invasiveness index was calculated based on the meta-analysis of data collected before PCV7 introduction across multiple countries (see 'Materials and methods'). A Spearman correlation test of the growth rate and prevalence of serotypes in carriage suggested a negative association between the variables. The association remained even after excluding serotype 3, which seemed like an outlier. (**c**) The relationship between the invasiveness and the average growth rates of each serotype from the pre-vaccination data. A Spearman correlation test of the growth rates and invasiveness suggested no association between the variables. All the error bars represent the 95% confidence intervals. Additional information is provided in *Supplementary file 3*.

## Average in vitro growth rates are not strongly correlated with a serotype's invasiveness or its prevalence in carriage or disease

Previous studies have elucidated that bacterial growth rate may influence colonisation and disease (*Small et al., 1986*), including systemic dissemination during infection (*Grant et al., 2009*) and disease manifestations (*Arends et al., 2022*; *Müller et al., 2020*). Pneumococcal strains that are prolific growers may exhibit better survival and replication in systemic tissues than those that are slow growers. We used isolates collected before the introduction of the PCV7 vaccine in the Netherlands (*Rodenburg et al., 2010*) to assess the association between in vitro growth rate of the isolates and their population-level characteristics: prevalence in invasive disease and nasopharyngeal carriage and invasiveness, that is, the odds of being isolated in invasive disease compared to the carriage state. Our first hypothesis was that the average in vitro growth rate for each pneumococcal serotype is positively associated with the prevalence of that serotype in invasive diseases. Our analysis revealed no evidence of an association between the prevalence of serotypes in invasive disease and the in vitro growth rate (Spearman correlation: –0.36, *P*=0.098) (*Figure 7a*, *Supplementary file 3*). Second, we hypothesised that the average in vitro growth rate of the serotypes is positively associated with the prevalence of the serotypes in the nasopharyngeal carriage, as fitter strains would grow faster than less fit strains. Similarly, we found no association between the prevalence of serotypes in carriage and the in vitro growth rate (Spearman correlation = −0.41, *P*=0.12) (*Figure 7b*, *Supplementary file 3*). We repeated the analysis after excluding serotype 3 and found no evidence of an association between growth rate and carriage prevalence (Spearman correlation = −0.28, *P*=0.314), which suggests that serotype 3 alone was driving the previously observed association. Finally, we tested the hypothesis that the growth rate is positively associated with the invasiveness of the serotypes. We found no association between the invasiveness and the in vitro growth rates of the serotypes (Spearman correlation: 0.144, *P*=0.594) (*Figure 7c*, *Supplementary file 3*). These findings suggest that growth rates alone may not be sufficient to explain the prevalence and invasiveness of pneumococcal serotypes, indicating the influence of other pneumococcal factors, microbiome (e.g., viral infection), and host-related factors in determining pneumococcal disease susceptibility.

## Discussion

Bacterial growth kinetics impact several traits, including gene and protein expression (*Klumpp et al., 2009*; *Nilsson et al., 1984*), antibiotic accumulation and efficacy (*Łapińska et al., 2022*; *Smirnova and Oktyabrsky, 2018*; *Tuomanen et al., 1986*), cell surface adhesion (*Rogers et al., 1984*), intra- and inter-specific competition (*Russell et al., 1979*), response to environmental conditions (*Hathaway et al., 2012*; *Tóthpál et al., 2019*; *Small et al., 1986*), plasmid replication (*Lin-Chao and Bremer, 1986*), bacteriophage dynamics (*Nabergoj et al., 2018*), colonisation and disease pathogenesis (*Im et al., 2022*), including clinical manifestation (*Im et al., 2022*; *Arends et al., 2022*; *Hathaway et al., 2012*). Whilst previous studies have attempted to understand bacterial, host, and environmental factors influencing bacterial growth, a major drawback has been a narrow focus on a small subset of strains (*Schaffner et al., 2014*). Therefore, the insights gained from these studies have been mostly limited to specific strains, as they ignored potential variability across different genetic backgrounds. Such population-level studies are critical for genetically diverse bacterial pathogens with highly structured populations (*Gladstone et al., 2019*), substantial antigenic diversity (*Bentley et al., 2006*), and recombination rates, such as the pneumococcus (*Donati et al., 2010*; *Croucher et al., 2013*; *Croucher et al., 2011*). Here, we investigate the association between pneumococcal genetics and growth kinetics using a combination of high-throughput quantification of in vitro pneumococcal growth, whole-genome sequencing, and population genomic analyses employing a bacterial GWAS approach. Notably, our findings reveal a strong influence of the capsular serotype compared to the genetic background, and the absence of specific individual genomic loci modulating pneumococcal growth kinetics, particularly the in vitro growth rate. These findings may partially explain why some serotypes and lineages colonise and cause invasive diseases more effectively than others and may have implications for developing measures to prevent and control pneumococcal infections.

Our findings suggest that the capsular serotype and genetic background influence the growth kinetics of pneumococci. The findings are generally consistent with results from experimental studies by others (*Arends et al., 2022*; *Hathaway et al., 2012*; *Tóthpál et al., 2019*), which used

capsule-switched strains to exclude the impact of the genetic background and demonstrated an alteration in growth dynamics due to the capsule. However, there may be potential differences in the growth media and conditions. Specifically, encapsulated strains exhibited higher growth rates than unencapsulated strains in nutrient-rich conditions (*Hathaway et al., 2012*). These findings suggest that the capsule is crucial for pneumococcal growth, but its effect depends on nutrient availability – promoting growth when nutrients are abundant and interfering with growth when nutrients are limited, respectively (*Hathaway et al., 2012*; *Müller et al., 2020*). A recent study by *Tóthpál et al., 2019* suggested that the serotype partially explains pneumococcal growth and suggested the genetic backgrounds, which vary in their ability to colonise the nasopharynx and cause invasive diseases (*Brueggemann et al., 2003*; *Hanage et al., 2005*; *Colijn et al., 2020*; *Massora et al., 2019*), may also play a role. However, the authors did not fully quantify the impact of the genetic background as the isolates were not sequenced (*Tóthpál et al., 2019*). Our study confirms the strong influence of both the pneumococcal capsular serotype and genetic background or lineage on the growth kinetics of the isolates. The influence of the serotype appeared to be greater than that of the genetic background on the average growth rate, while the other growth features seemed to show an opposite trend. These findings suggest that the respective influence of the capsular serotype and lineage varies by specific derived pneumococcal growth features.

Interestingly, although our findings revealed that the capsular serotype and genetic background influence pneumococcal growth kinetics, statistical analysis showed that the phenotypic variability attributed to the capsular serotype was greater than that attributed to the genetic background. Specifically, the capsular serotype exhibits a more pronounced growth rate, maximum growth change, and maximum growth density than the genetic background. In contrast, the genetic background had a greater effect on the lag phase duration than the serotype, a growth phase suggested to be responsible for bacterial adaptation to new environments (*Rolfe et al., 2012*). We acknowledge that our comparison of growth features in the same genetic background may also indicate a significant amount of genetic variation for some of the lineages, some of which could be responsible for changes in growth kinetics. These findings illustrate the combined, albeit variable, importance of the capsular serotype and genetic background on pneumococcal growth kinetics, which may partially explain the pathogenesis and prevalence of the pneumococcus in nasopharyngeal carriage and invasive disease.

Previous studies have revealed individual bacterial genomic loci that impact other phenotypes, including pathogenicity (*Lees et al., 2019a*; *Li et al., 2019b*; *Young et al., 2019*; *Chaguza et al., 2022*; *Davies et al., 2019*; *Chewapreecha et al., 2019*; *Peters et al., 2021*; *Biggel et al., 2020*; *Wee et al., 2021*), virulence (*Laabei et al., 2014*; *Chaguza et al., 2020*; *Boeck et al., 2022*; *Galardini et al., 2020*), host, niche, and environmental adaptation (*Chaguza et al., 2022*; *Chewapreecha et al., 2019*; *Ma et al., 2020*), conjugation (*Clark et al., 2022*), and antimicrobial resistance (*Chewapreecha et al., 2014*; *Coll et al., 2018*; *Creasy-Marrazzo et al., 2022*; *Farhat et al., 2013*; *Hicks et al., 2019*; *Farhat et al., 2019*; *Alam et al., 2014*; *Mortimer et al., 2022*), independent of the genetic background. Although our study focused primarily on the pneumococcal growth rate parameter, the other growth parameters are also biologically relevant. For example, the lag phase duration potentially reflects the ability to adapt to environmental conditions, such as nutrient availability or stress, and may be more influenced by regulatory genes involved in sensing and responding to environmental cues (*Rolfe et al., 2012*; *Bertrand, 2019*). On the other hand, the average growth rate may be impacted by genes controlling the rate of cell division under optimal conditions (*Li et al., 2019a*) while the maximum growth density, which reflects the final cell density, might be shaped by factors related to nutrient utilisation efficiency, waste tolerance, or quorum sensing. The duration of the stationary phase may reflect the initiation of the lytic growth phase, possibly due to the effects of an autolysin inhibitor (*Höltje and Tomasz, 1975*). Our GWAS analysis, undertaken to gain a more nuanced understanding of how genetic variants contribute to different physiological aspects of microbial growth, revealed no elevated signals in specific genomic loci associated with pneumococcal growth features. This suggests that individual variants may not significantly influence pneumococcal growth kinetics independently of the genetic background and serotype. Previous studies have shown that mutations in other capsule biosynthesis genes (*wchA* or *cpsE*) encoding a protein that links activated glucose phosphates to lipid carriers (*Bentley et al., 2006*) also influence the growth dynamics of the pneumococcus (*Schaffner et al., 2014*). However, our results contradict our initial speculation that genetic variation in the capsule polysaccharide synthesis genes would be strongly associated with

pneumococcal growth features. These findings suggest that the contribution of specific individual pneumococcal genomic loci to growth kinetics does not stand out, hinting at the potential polygenic nature of growth kinetics and an overall impact of the genetic background, which exerts a cumulative effect, diluting the influence of individual loci. Uncovering such effects will require follow-up studies with larger sample sizes.

Compared to the findings of a previous study by *Hathaway et al., 2012*, we found no robust evidence for an association between the in vitro growth rates and the population-level prevalence of the serotypes in nasopharyngeal carriage. However, this may have occurred as a result of a limitation of our study, since we studied a collection of randomly sampled invasive isolates, and the growth experiments were conducted under conditions similar to those of blood. In contrast, *Hathaway et al., 2012* used a non-random selection of both carriage and disease isolates. Isolates likely have adaptations to the environmental conditions in which they were isolated; therefore, in vitro growth characteristics for any given isolate may be different under experimental conditions simulating different niches. In addition, the in vitro growth environment cannot fully represent in vivo conditions and fails to consider the impact of selective pressures arising from serotype-specific responses (antibodies) or non-specific responses (phagocytosis) as well as potential interactions with other pathogens. The lack of association between the invasiveness and prevalence of serotypes in invasive disease and carriage, as well as the growth rates, could reflect that the conditions in the in vitro growth environment do not fully represent in vivo conditions. Collectively, these findings suggest that these and other factors may substantially influence the complex pathogenesis of pneumococcal disease in vivo.

The strength of this study lies in the analysis of an extensive collection of well-phenotyped natural isolates, combined with phenotyping and the estimation of growth features using statistical models and statistical genetics to determine the association between the growth features and genetic variation across the entire genome. As previously noted by *Tóthpál et al., 2019*, the culturing history of the isolates may impact the growth kinetics. In our study, we consistently cultured all isolates after isolation from patients using standardised culturing procedures (single assay), identical initial concentration, environmental conditions (temperature), and equipment to measure optical density, and repeated measurements to spot technical variability (*Arends et al., 2022*). Compared to the present study, future studies would benefit from larger sample sizes and may yield more results, as we only explained a portion of the heritability of the growth features. We also acknowledge that the methods used to estimate heritability may also sometimes perform poorly on certain bacterial species; therefore, more robust methods are needed to validate our findings further (*Mallawaarachchi et al., 2022*). We conducted our in vitro culture experiments for 15 hours, as the growth curves indicate that most isolates had reached the stationary phase by that time. Increasing the culturing time from 15 to >24 hours, as done elsewhere (*Hathaway et al., 2016*; *Hamaguchi et al., 2018*), might be considered for follow-up studies. However, *Streptococcus pneumoniae* undergoes autolysis upon reaching a specific cell density, which could distort growth measurements and complicate interpretation if incubation were prolonged. Furthermore, since the pneumococcus invades different niches and systemic tissues under varying conditions, including oxygen concentration and temperature (*Im et al., 2022*; *Hathaway et al., 2012*; *Tóthpál et al., 2019*; *Small et al., 1986*), future studies should investigate the effect of these environmental conditions on pneumococcal growth and identify pneumococcal genomic variants associated with growth kinetics for each condition. Considering that our dataset contained only a third of the >100 known pneumococcal serotypes globally (*Ganaie et al., 2020*), due to the geographical variation in the distribution of pneumococcal serotypes (*Johnson et al., 2010*), future studies should investigate all serotypes to provide further insights into the impact of different serotypes on pneumococcal growth kinetics.

Additionally, considering that we generated the phylogeny of the entire pneumococcal species rather than isolates belonging to a single strain or lineage, as defined by several approaches (*Gladstone et al., 2019*; *McGee et al., 2001*; *Cheng et al., 2013*), we did not remove recombinations using widely used methods designed for clonal bacterial populations, such as Gubbins (*Croucher et al., 2015b*). Therefore, there is a possibility that some branch lengths in the phylogeny used for ancestral trait reconstruction analysis may be less accurate, although the topology is usually robust to recombination (*Hedge and Wilson, 2014*). Furthermore, there may be other variants that potentially influence the pneumococcal growth kinetics, which are regarded as not statistically significant in this study based on our conservative genome-wide significance threshold. As more similar studies are

conducted, particularly those utilising much larger datasets, potential variants associated with pneumococcal growth kinetics may be discovered.

Our analysis of population-level genomic and high-throughput in vitro pneumococcal growth data suggests that the capsular serotype and lineage influence pneumococcal growth. However, although some growth features are highly heritable, there appears to be no specific genetic variants that strongly influence these phenotypes, suggesting that a combination of genomic loci with minor effects, individually or collectively, may play a greater role in pneumococcal growth kinetics. Overall, our study provides a proof-of-concept for combining phenotyping and population genomics to understand the genetic basis for pneumococcal growth, an approach which can be applied to study other bacteria of clinical significance and those used in bioengineering, genetics, and the food industry (*Tonner et al., 2020*; *Yang et al., 2018*; *Alkema et al., 2016*). Ultimately, we anticipate that applying our approach will have implications for devising strategies to reduce life-threatening bacterial infections.

## Materials and methods
### Sample characteristics and whole-genome sequencing
In total, 348 (n=348) pneumococcal isolates collected from patients with invasive pneumococcal disease through the PBCN observational cohort study (*Cremers et al., 2014*) with in vitro growth data were available for the study. We included a single isolate per patient as invasive pneumococcal infections are typically clonal, so a single isolate is sufficient. The consecutive blood isolates were collected at two hospitals in Nijmegen, the Netherlands, before (2000–2006) and after (2007–2011) the introduction of the PCV7 vaccine into the Dutch paediatric immunisation programme (*Cremers et al., 2014*). Most of the bacteraemia cases (97%) concerned adults. Additional information on the PBCN study, including collected clinical variables and phenotypic in vitro growth data, has been previously described (*Arends et al., 2022*; *Cremers et al., 2014*; *Cremers et al., 2019*; *Cremers et al., 2015*). To generate whole-genome sequences, we sequenced a single isolate per patient using an Illumina HiSeq sequencer (Illumina, San Diego, CA, USA). We then assembled the generated reads using SPAdes genome assembler (version 3.14.0) (*Bankevich et al., 2012*), and evaluated the draft assemblies using assembly-stats (version: 1.0.1) (https://github.com/sanger-pathogens/assembly-stats) (RRID:SCR_023963). In addition, we used a combination of metrics, including genome size and the number of contigs, to assess the quality of the draft assemblies for inclusion in the analysis.

### Measurement of in vitro pneumococcal growth kinetics and derivation of growth features
We measured the pneumococcal growth for slightly over 15 hours as described by *Arends et al., 2022* In brief, we incubated a frozen aliquot of standardised pneumococcal inoculum in 1.5 ml of blood-like medium (50% M17, 50% casamino acids tryptone [CAT] medium, 0.25% glucose). We measured growth kinetics at 37°C and 5% $CO_2$ every 10 minutes for over 15 hours at an $OD_{620}$ using a humidity cassette with a microplate reader (Spark 10M, Tecan, Switzerland). We added 15 μL of inoculum was added to 1.5 mL of prewarmed rich growth medium (45% M17, 45% CAT, 0.225% glucose, 10% fetal calf serum [Greiner Bio-one], 26 U/mL catalase [Sigma-Aldrich C1345]) in each well of a sterile flat-bottomed 48-well plate (Nunclon Surface, Nunc, Denmark). The growth medium was supplemented with catalase. We selected these nutrient-rich conditions to highlight the differences in the growth features of the pneumococcus. We performed six repeat measurements on three separate days for each isolate. To understand the growth kinetics of the invasive Dutch pneumococcal isolates, we fitted a modified logistic growth curve function, that is, $L_0 + L_{max}/\left(1 + e^{-r(t-t_0)}\right)$, separately to the growth data for each isolate and replicate, using the 'optim' function in stats (version 4.3.2) R package (https://www.R-project.org/). We then averaged the inferred values for each replicate to obtain average estimates for each parameter. The parameter $L_0$ represents the initial cell density measured as absorbance or optical density at a wavelength of 620 nm ($OD_{620}$); $L_{max}$ is the maximum cell density; $t_0$ is the initial time, and $r$ is the average growth rate. We fitted two curves in total for each isolate based on the averaged data from the six replicates using the limited-memory Broyden–Fletcher–Goldfarb–Shanno (L-BFGS-B) optimisation algorithm by minimising the residual sum of squares (*Byrd et al., 1995*). We fitted the first curve to all the data points and the second curve to the data measured until the end of the stationary phase or the beginning of the death phase, whichever came

earlier. We defined the end of the stationary phase or the start of the death phase as the time when the optical cell density decreased by more than 5% from the maximum cell density. The curve fitted based on the data measurements collected up to the end of the stationary phase or death phase, or the end of the experiment, if none of these phases had been reached, captured the growth dynamics better than the curve fitted using all the data points to accurately estimate the growth rate and the maximum cell density corresponding to the peak of the pneumococcal growth curve, especially for growth curves with observably short stationary phases and rapid autolysis. Therefore, parameters for the downstream analyses were inferred from the curves fitted to the reduced dataset.

We determined four growth phenotypes for each pneumococcal isolate estimated from the fitted growth curve, namely the lag-phase duration, maximum growth density ($H_{max}$), that is, captured by parameter $L_{max}$, maximum growth change ($\Delta H$) corresponding to the difference of the maximum and minimum density ($L_{max}$-$L_0$), and average growth rate ($r$). We defined the lag-phase duration as the time when the optical cell density was at least 1.5-fold higher than the initial cell density at the start of the experiment, as described by *Arends et al., 2022*. Due to the challenges of characterising the kinetics at the stationary or death phase, partly due to the high strain-to-strain variability or short study period, we were unable to derive the growth features that capture these phases, including the duration of the stationary and death phases. The logistic growth curves showed an excellent fit for the data, providing an opportunity to investigate the impact of capsular serotype and genetic background (or lineage), as well as specific genomic loci, independent of the genetic background, on the estimated growth parameters and features.

## Molecular typing of sequenced isolates

Molecular typing of the isolates to determine capsular serotypes and sequence types (STs) was done in silico using Pathogenwatch (https://pathogen.watch/), a global platform for genomic surveillance (*Argimón et al., 2021*). We determined the capsular serotypes and STs using SeroBA (version v1.0.1) (*Epping et al., 2018*) and the MLST scheme for the pneumococcus (*Enright and Spratt, 1998*), respectively. Pneumococcal lineages were defined based on the GPSC nomenclature (*Gladstone et al., 2019*) using PopPUNK (version 1.1.0) (*Lees et al., 2019b*). Additionally, we checked the species of each sequenced isolate using Speciator (version 3.0.1) in Pathogenwatch.

## Construction of multiple sequence alignments and phylogenetic trees

We mapped the generated sequence reads for each isolate and a draft assembly of *Streptococcus oralis* strain SF100 genome (GenBank accession: NZ_CP069427) to the pneumococcal ATCC 700669 reference genome (GenBank accession: NC_011900) using Snippy (version 4.6.0) (https://github.com/tseemann/snippy) (RRID:SCR_023572). We identified 155,305 SNPs in the generated alignment using SNP-sites (version 2.3.2) (*Page et al., 2016*) and then used them to construct a maximum-likelihood phylogeny using IQ-TREE (version 2.0.3) (*Nguyen et al., 2015*). We used the alignment of variable sites by mapping to a reference genome to generate a maximum-likelihood phylogeny using IQ-TREE, as it has been shown to produce more accurate phylogenies (*Lees et al., 2018b*). The general time-reversible (GTR) nucleotide substitution model and Gamma heterogeneity of the substitution rates among sites. We assessed branch support using the ultrafast bootstrap approximation specified by the "-B 1000" option (*Hoang et al., 2018*). We optimised bootstrapped trees using nearest neighbour interchange method on bootstrap alignments by selecting the "--bnni" option. We rooted the phylogenetic tree of the pneumococcal isolates using *S. oralis* strain SF100 genome as an outgroup (not shown in the phylogeny). An annotated and interactive version of the phylogeny is available on the Microreact web tool (*Argimón et al., 2016*) (https://microreact.org/project/wxmdblfgprepwugvbbtaq7).

## Quantifying the phylogenetic signal

To assess the correlation between the estimated growth features and pneumococcal phylogeny, we used models of continuous character evolution to estimate the phylogenetic signal using Pagel's $\lambda$ (*Pagel, 1999*) implemented in the 'phylosig' function in phytools package (version 2.1.1) (*Revell, 2012*). Pagel's $\lambda$ transforms the internal branches relative to the terminal branches and generates values typically ranging from 0 to 1, with smaller values indicating no phylogenetic signal or independent evolution of the phenotype from the phylogeny. In comparison, higher values close to one show

a high phylogenetic signal or a decreased phenotype variability among genetically similar taxa at the tips of the phylogeny. First, we transformed the estimated growth features to a normal distribution using a rank-based inverse normal transform implemented in the RNOmni package (version 1.0.1.2) (https://cran.r-project.org/package=RNOmni). Next, we mapped the estimated continuous growth feature values onto the phylogeny using fast estimation of maximum likelihood for ancestral states (*Revell, 2012*).

## Bacterial GWAS

We identified the presence and absence patterns of unitig sequences to capture genetic variation in the pneumococcal genomes included in this study for the GWAS analysis. We identified unitig sequences from 31bp *k*-mers using Bifrost (version 1.0.6) (*Holley and Melsted, 2020*). First, we inferred unitig sequences from a compacted De Bruijn graph constructed using *k*-mers of length 31, based on draft assemblies of all the pneumococcal isolates included in this study. We generated De Bruijn graphs for each isolate separately and then queried them with the unitig sequences found in the entire dataset to identify the presence and absence of each unitig found in each sequenced genome. We considered a unitig present when 100% of the *k*-mer was presented in the De Bruijn graph. We combined the presence and absence patterns of the unitig sequences from all the isolates to construct a single matrix.

Since the linear mixed effects models require the tested continuous phenotype to be normally distributed, we used the normalised growth features after applying the rank-based inverse normal transformation using the RNOmni package (version 1.0.1.2) (https://www.R-project.org/) We performed GWAS analysis of pneumococcal growth features using robust linear mixed-effects models implemented in Pyseer (version 1.3.7-dev) (*Lees et al., 2018a*). A kinship matrix specifying the variances and covariances of the random effects for each isolate was generated using the 'similarity_ pyseer' script in Pyseer. Using this kinship matrix, Pyseer fits an unobserved random effect for each isolate to account for the clonal population structure in the GWAS. We excluded unitigs with <5% minimum allele frequency and >95% maximum allele frequency. The inferred statistical significance for each variant was adjusted to correct for multiple testing using the Bonferroni correction. We estimated the total number of unique unitig presence and absence patterns in the dataset using a script included in Pyseer (https://github.com/mgalardini/pyseer/blob/master/scripts/count_patterns.py) (*Galardini and Lees, 2025*). We used the obtained value when adjusting for multiple testing using the Bonferroni correction approach. Therefore, we defined the genome-wide statistical significance threshold as $P<2.58 \times 10^{-07}$, that is, α/G, where α=0.05 and G=194,053 is the number of unique unitig presence and absence patterns, while the suggestive threshold is $P<5.15 \times 10^{-06}$, whereby α=1. To check potential issues arising from the population structure, we generated QQ plots from the output of GWAS analysis for each growth feature to compare the expected and observed *P*-values using qqman (version 0.1.7) (*Turner, 2014*). The overall proportion of phenotypic variability explained by variation in the genome (narrow-sense heritability [$h^2$]) was estimated using GEMMA (version 0.98.1) (*Zhou and Stephens, 2012*) and Pyseer. The input files for GEMMA were prepared based on the unitig presence and absence data using PLINK (version 1.90b4) (*Purcell et al., 2007*). To identify variants associated with the growth features independently of the serotype, we repeated the GWAS by including the serotype as a covariate.

## Annotation of variants identified by the GWAS analysis

To annotate the unitig sequences with statistical significance values below the suggestive and genome-wide significance thresholds, we used a custom BioPython (*Cock et al., 2009*) wrapper script for BLASTN (version 2.5.0+) (*Altschul et al., 1990*) to identify genomic features in the ATCC 700669 reference genome (GenBank accession: NC_011900) and other complete pneumococcal genomes downloaded from GenBank (*Benson et al., 2008*). The GWAS results were summarised and visualised using Manhattan plots in R (version 4.0.3) [https://www.R-project.org/].

## Other statistical analysis

We compared the growth features between serotypes and lineages using the Kruskal–Wallis non-parametric test. The distribution of the growth features was plotted using violin plots using the vioplot package (version 0.4.0) (https://CRAN.R-project.org/package=vioplot). To assess the contribution of

serotype and lineage to the variability in the pneumococcal growth features, we used linear mixed-effects models implemented in the lme4 package (version 1.1.35.1) (*Bates et al., 2015*), with each growth feature specified as the outcome and the serotype and lineage as random effects. The statistical significance for the linear mixed model was inferred using the lmerTest package (version 3.1.3) (https://CRAN.R-project.org/package=lmerTest). To assess the contribution of serotype and lineage to the variability of each growth feature, we calculated the proportion of variance attributable to the serotype and lineage components of the total variance. We calculated the average invasiveness of the serotypes based on a meta-analysis of data collected in different countries in infants before the introduction of the PCVs (*Colijn et al., 2020*) using the random-effects model implemented in the metafor package (version 2.4.0) (https://CRAN.R-project.org/package=metafor). We defined invasiveness as the odds ratio of detecting a pneumococcal serotype among isolates causing invasive disease compared to nasopharyngeal carriage. We added a value of 1 to cells with zero counts. Due to the unavailability of the nasopharyngeal carriage data, we imputed the prevalence of serotypes in the carriage by dividing the prevalence in invasive disease by the average invasiveness of each serotype. Next, we used a Spearman correlation test to assess the association between the mean prevalence of serotypes in the nasopharyngeal carriage and invasive disease, and average invasiveness with the average in vitro pneumococcal growth rate.

## Acknowledgements

We would like to thank the study participants, guardians, and the clinical and laboratory staff who collected and processed the samples at various laboratories in the Netherlands. We would also like to acknowledge the support of the sequencing, core, parasites and microbes programme, the Bentley lab at the Wellcome Sanger Institute, and the Weinberger and Pitzer labs at the Yale School of Public Health for their feedback on the analysis. We acknowledge the participating hospitals and affiliated researchers for their contributions and support. This study was supported by the TARGET project (grant number: JPIAMR2019-087) funded by JPI-AMR-ZonMW, the Netherlands, and by the SNSF Swiss Postdoctoral Fellowship (SNSF grant number: TMPFP3_209768). Activities at the Wellcome Sanger Institute were funded by the Bill and Melinda Gates Foundation (grant number: OPP1034556) and Wellcome Trust (2016–2021 core award grant number: 206194). The funders had no role in study design, data collection, analysis, decision to publish, or manuscript preparation. The findings do not necessarily reflect the official views and policies of the author's institutions and funders. For the purpose of open access, the authors have applied a CC BY public copyright licence to any Author Accepted Manuscript version arising from this submission.

## Additional information

### Funding

| Funder | Grant reference number | Author |
| --- | --- | --- |
| JPI-AMR-ZonMW | JPIAMR2019-087 | Marien I de Jonge |
| Swiss National Science Foundation | TMPFP3_209768 | Amelieke JH Cremers |
| Gates Foundation | OPP1034556 | Stephen D Bentley |
| Wellcome Trust | 10.35802/206194 | Stephen D Bentley |

The funders had no role in study design, data collection and interpretation, or the decision to submit the work for publication. For the purpose of Open Access, the authors have applied a CC BY public copyright license to any Author Accepted Manuscript version arising from this submission.

### Author contributions

Chrispin Chaguza, Conceptualization, Data curation, Formal analysis, Validation, Visualization, Methodology, Writing – original draft, Writing – review and editing; Daan W Arends, Indri Hapsari Putri, Data curation, Investigation, Methodology, Writing – review and editing; Stephanie W Lo, Data

curation, Funding acquisition, Writing – review and editing; Anna York, Anne L Wyllie, Writing – review and editing; John A Lees, Software, Supervision, Methodology, Writing – review and editing; Daniel M Weinberger, Conceptualization, Resources, Supervision, Investigation, Methodology, Writing – original draft, Project administration, Writing – review and editing; Stephen D Bentley, Conceptualization, Resources, Software, Supervision, Funding acquisition, Investigation, Methodology, Writing – original draft, Project administration, Writing – review and editing; Marien I de Jonge, Conceptualization, Resources, Data curation, Supervision, Funding acquisition, Investigation, Methodology, Writing – original draft, Project administration, Writing – review and editing; Amelieke JH Cremers, Conceptualization, Resources, Data curation, Supervision, Funding acquisition, Validation, Investigation, Methodology, Writing – original draft, Project administration, Writing – review and editing

## Author ORCIDs
Chrispin Chaguza ⓘ https://orcid.org/0000-0002-2108-1757
Anne L Wyllie ⓘ https://orcid.org/0000-0001-6015-0279
Daniel M Weinberger ⓘ https://orcid.org/0000-0003-1178-8086
Amelieke JH Cremers ⓘ https://orcid.org/0000-0003-4704-4978

## Ethics

Human subjects: The isolates used in this work were collected through an observational cohort study Pneumococcal Bacteraemia Collection Nijmegen (PBCN). The non-applicability of the Medical Research Involving Human Subjects Act (WMO) on the study design was confirmed by the Regional Ethics Committee, and the local medical ethics committees of the participating hospitals in the Netherlands approved the study procedures. Study procedures were approved by the Medical Ethical committees of the participating hospitals, including a waiver for individual informed consent (file number: 2020-6644 Radboudumc).

Reviewer #1 (Public review): https://doi.org/10.7554/eLife.105555.3.sa1
Reviewer #2 (Public review): https://doi.org/10.7554/eLife.105555.3.sa2
Reviewer #3 (Public review): https://doi.org/10.7554/eLife.105555.3.sa3
Author response https://doi.org/10.7554/eLife.105555.3.sa4

# Additional files

## Supplementary files
Supplementary file 1. Characteristics of the pneumococcal isolates used in this study and summary of the derived growth features or parameters.

Supplementary file 2. Summary of the phylogenetic signal and narrow-sense heritability of the growth features.

Supplementary file 3. Summary of the serotype prevalence and growth rates used.

MDAR checklist

## Data availability

We have deposited the whole-genome sequencing data for the study isolates in the European Nucleotide Archive (ENA) and provided the accession numbers and other isolate metadata in *Supplementary file 1*. All the other data supporting the findings of this study are described in this paper or are available as part of the supplementary material. We have also provided additional raw data and code used for the analysis on GitHub: https://github.com/ChrispinChaguza/SpnGrowthKinetics (copy archived at *Chaguza, 2025*).

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
