## [Editor Report · eLife Assessment]

This is an **important** study that examines the impact of *Streptococcus pneumoniae* genetics on its in vitro growth kinetics, aiming to identify potential targets for vaccines and therapeutics. The study identified significant variations in growth characteristics among capsular serotypes and lineages, linked to phylogeny and high heritability, but genome-wide association studies did not reveal specific genomic loci associated with growth features independent of the genetic background. The evidence supporting these findings is **convincing**.

---

## [Referee Report · Reviewer #1 (Public review)]

Summary:

This manuscript uses a diverse isolate collection of Streptococcus pneumoniae from hospital patients in the Netherlands to understand the population-level genetic basis of growth rate variation in this pathogen, which is a key determinant of S. pneumoniae within-host fitness. Previous efforts have studied this phenomenon in strain-specific comparisons, which can lack the statistical power and scope of population-level studies. The authors collected a rigorous set of in vitro growth data for each S. pneumoniae isolate and subsequently paired growth curve analysis with whole-genome analyses to identify how phylogenetics, serotype and specific genetic loci influence in vitro growth. While there were noticeable correlations between capsular serotype and phylogeny with growth metrics, they did not identify specific loci associated with altered in vitro growth, suggesting that these phenotypes are controlled by the collective effect of the entire genetic background of a strain. This is an important finding that lays the foundation for additional, more highly-powered studies that capture more S. pneumoniae genetic diversity to identify these genetic contributions.

Strengths:

The authors were able to completely control the experimental and genetic analyses to ensure all isolates underwent the same analysis pipeline to enhance the rigor of their findings.

The isolate collection captures an appreciable amount of S. pneumoniae diversity and, importantly, enables disentangling the contributions of the capsule and phylogenetic background to growth rates.

This study provides a population-level, rather than strain-specific, view of how genetic background influences growth rate in S. pneumoniae. This is an advance over previous studies that have only looked at smaller sets of strains.

The methods used are well-detailed and robust to allow replication and extension of these analyses. Moreover, the manuscript is very well written and includes a thoughtful and thorough discussion of the strengths and limitations of the current study.

Weaknesses:

As acknowledged by the authors, the genetic diversity and sample size of this newly collected isolate set is still limited relative to the known global diversity of S. pneumoniae, which evidently limits the power to detect loci with smaller/combinatorial contributions to growth rate (and ultimately infection).

The in vitro growth data is limited to a single type of rich growth medium, which may not fully reflect the nutritional and/or selective pressures present in the host.

The current study does not use genetic manipulation or in vitro/in vivo infection models to experimentally test whether alteration of growth rates as observed in this study is linked to virulence or successful infection. The availability of a naturally diverse collection with phylogenetic and serotype combinations already identified as interesting by the authors provides a strong rationale for wet-lab studies of these phenotypes.

Update on first revision:

The authors have responded to all of my initial comments as well as those of the other reviewers, and I have no further concerns to be addressed.

---

## [Referee Report · Reviewer #2 (Public review)]

The study by Chaguza et al. presents a novel perspective on pneumococcal growth kinetics, suggesting that the overall genetic background of Streptococcus pneumoniae, rather than specific loci, plays a more dominant role in determining growth dynamics. Through a genome-wide association study (GWAS) approach, the authors propose a shift in how we understand growth regulation, differing from earlier findings that pinpointed individual genes, such as wchA or cpsE, as key regulators of growth kinetics. This study highlights the importance of considering the cumulative impact of the entire genetic background rather than focusing solely on individual genetic loci.

The study emphasizes the cumulative effects of genetic variants, each contributing small individual impacts, as the key drivers of pneumococcal growth. This polygenic model moves away from the traditional focus on single-gene influences. Through rigorous statistical analyses, the authors persuasively advocate for a more holistic approach to understanding bacterial growth regulation, highlighting the complex interplay of genetic factors across the entire genome. Their findings open new avenues for investigating the intricate mechanisms underlying bacterial growth and adaptation, providing fresh insights into bacterial pathogenesis.

Strengths:

This study exemplifies a holistic approach to unraveling key factors in bacterial pathogenesis. By analyzing a large dataset of whole-genome sequences and employing robust statistical methodologies, the authors provide strong evidence to support their main findings. Which is a leap forward from previous studies focused on a relatively smaller number of strains. Their integration of genome-wide association studies (GWAS) highlights the cumulative, polygenic influences on pneumococcal growth kinetics, challenging the traditional focus on individual loci. This comprehensive strategy not only advances our understanding of bacterial growth regulation but also establishes a foundation for future research into the genetic underpinnings of bacterial pathogenesis and adaptation. The amount of data generated and corresponding approaches to analyze the data are impressive as well as convincing. The figures are convincing and comprehensible too. The revised version of the manuscript, after the addition and including explanations, is more convincing and acceptable.

Weaknesses:

This study suggests evidence that the genetic background significantly influences bacterial growth kinetics. However, the absence of experimental validation remains a critical limitation. Although the authors acknowledge in their response to reviewers that bench-experiments were beyond the scope of this work and are planned, this gap of experimental validation weakens the current conclusions. Demonstrable validation will be essential to corroborate the associations identified through the GWAS approach. Future experimental efforts will be critical to substantiate these findings and to deepen our understanding of the genetic determinants governing bacterial growth dynamics.

---

## [Referee Report · Reviewer #3 (Public review)]

This study provides insights into the growth kinetics of a diverse collection of Streptococcus pneumoniae, identifying capsule and lineage differences. It was not able to identify any specific loci from the GWAS that were associated with the growth features. It does provide a useful study linking phenotypic data with large scale genomic population data.

In the revised version, the authors have addressed the points raised by the reviewers. The authors have provided additional detail in the Introduction and Methods that both improves the general accessibility for the broad readership of eLife, and the ability of other researchers to reproduce the approaches used in this study. They have expanded the Results and Discussion text in some sections to provide greater clarity and accuracy in reporting their data.

The inclusion of a Data Availability statement was a useful addition and will help ensure the manuscript adheres to eLife's publishing policies.

---

## [Author Response]

The following is the authors’ response to the original reviews

**Reviewer #1 (Public review):**
Summary:This manuscript uses a diverse isolate collection of Streptococcus pneumoniae from hospital patients in the Netherlands to understand the population-level genetic basis of growth rate variation in this pathogen, which is a key determinant of S. pneumoniae within-host fitness. Previous efforts have studied this phenomenon in strain-specific comparisons, which can lack the statistical power and scope of population-level studies. The authors collected a rigorous set of in vitro growth data for each S. pneumoniae isolate and subsequently paired growth curve analysis with whole-genome analyses to identify how phylogenetics, serotype, and specific genetic loci influence in vitro growth. While there were noticeable correlations between capsular serotype and phylogeny with growth metrics, they did not identify specific loci associated with altered in vitro growth, suggesting that these phenotypes are controlled by the collective effect of the entire genetic background of a strain. This is an important finding that lays the foundation for additional, more highly-powered studies that capture more S. pneumoniae genetic diversity to identify these genetic contributions.

Thank you for an excellent summary of our manuscript.

Strengths:(1) The authors were able to completely control the experimental and genetic analyses to ensure all isolates underwent the same analysis pipeline to enhance the rigor of their findings.(2) The isolate collection captures an appreciable amount of S. pneumoniae diversity and, importantly, enables disentangling the contributions of the capsule and phylogenetic background to growth rates.(3) This study provides a population-level, rather than strain-specific, view of how genetic background influences the growth rate in S. pneumoniae. This is an advance over previous studies that have only looked at smaller sets of strains.(4) The methods used are well-detailed and robust to allow replication and extension of these analyses. Moreover, the manuscript is very well written and includes a thoughtful and thorough discussion of the strengths and limitations of the current study.

Thank you for excellently summarising the strengths of our manuscript.

Weaknesses:(1) As acknowledged by the authors, the genetic diversity and sample size of this newly collected isolate set are still limited relative to the known global diversity of S. pneumoniae, which evidently limits the power to detect loci with smaller/combinatorial contributions to growth rate (and ultimately infection).

Indeed, while larger pneumococcal datasets exist globally, most of these datasets do not have reliable metadata on in vitro growth rates and other phenotypes, as the intention, for the most part, is to conduct population-level surveillance to track the changes in the serotype distribution to assess the impact of introducing pneumococcal conjugate vaccines. In this study, we adopted a different approach to phenotypically characterising the samples collected from these surveillance studies to understand the genetic features that influence the intrinsic growth characteristics of the isolates. While our dataset size is modest, it exemplifies how we can combine whole-genome sequencing and phenotypic characterisation of bacterial isolates to understand the genetic determinants that may drive intrinsic phenotypic differences between strains.

(2) The in vitro growth data is limited to a single type of rich growth medium, which may not fully reflect the nutritional and/or selective pressures present in the host.

We agree that our study focused on a single type of rich growth medium, which may not fully reflect the nutritional or selective pressures present in the host. The rationale and the representativeness of the selected culture conditions were more extensively discussed in Arends et al. (10.1128/spectrum.00050-22). Considering that this was a proof-of-concept study to assess the feasibility of our approach, future studies by us and others will evaluate the impact of using different media. Besides the media, complementary techniques such as transcriptome sequencing will help uncover additional insights into potential factors that influence differences in pneumococcal growth kinetics.

(3) The current study does not use genetic manipulation or in vitro/in vivo infection models to experimentally test whether alteration of growth rates as observed in this study is linked to virulence or successful infection. The availability of a naturally diverse collection with phylogenetic and serotype combinations already identified as interesting by the authors provides a strong rationale for wet-lab studies of these phenotypes.

We concur that additional genetic manipulation studies to assess the impact of altering growth rates on virulence and infection would have provided further insights. While this was beyond the scope of this study, we plan to conduct follow-up work to assess this using carefully selected strains from our pneumococcal collection. Because our current study demonstrates that genetic determinants of pneumococcal growth features are not simply confined to single loci, such experimental validation would require novel wet-lab approaches that consider epistatic interactions. In addition, in vivo infection models that allow the study of dissemination from the bloodstream are not yet well established.

**Reviewer #2 (Public review):**
Summary:The study by Chaguza et al. presents a novel perspective on pneumococcal growth kinetics, suggesting that the overall genetic background of Streptococcus pneumoniae, rather than specific loci, plays a more dominant role in determining growth dynamics. Through a genome-wide association study (GWAS) approach, the authors propose a shift in how we understand growth regulation, differing from earlier findings that pinpointed individual genes, such as wchA or cpsE, as key regulators of growth kinetics. This study highlights the importance of considering the cumulative impact of the entire genetic background rather than focusing solely on individual genetic loci.The study emphasizes the cumulative effects of genetic variants, each contributing small individual impacts, as the key drivers of pneumococcal growth. This polygenic model moves away from the traditional focus on single-gene influences. Through rigorous statistical analyses, the authors persuasively advocate for a more holistic approach to understanding bacterial growth regulation, highlighting the complex interplay of genetic factors across the entire genome. Their findings open new avenues for investigating the intricate mechanisms underlying bacterial growth and adaptation, providing fresh insights into bacterial pathogenesis.

Thank you for an excellent summary of our manuscript.

Strengths:This study exemplifies a holistic approach to unraveling key factors in bacterial pathogenesis. By analyzing a large dataset of whole-genome sequences and employing robust statistical methodologies, the authors provide strong evidence to support their main findings. Which is a leap forward from previous studies focused on a relatively smaller number of strains. Their integration of genome-wide association studies (GWAS) highlights the cumulative, polygenic influences on pneumococcal growth kinetics, challenging the traditional focus on individual loci. This comprehensive strategy not only advances our understanding of bacterial growth regulation but also establishes a foundation for future research into the genetic underpinnings of bacterial pathogenesis and adaptation. The amount of data generated and corresponding approaches to analyze the data are impressive as well as convincing. The figures are convincing and comprehensible too.

Thank you for pointing out the strengths of our manuscript excellently.

Weaknesses:Despite the strong outcomes of the GWAS approach, this study leaves room for differing interpretations. A key point of contention lies in the title, which initially gives the impression that the research addresses growth kinetics under both in vitro and in vivo conditions. However, the study is limited to in vitro growth kinetics, with the assumption that these findings are equally applicable to in vivo scenarios-a premise that is not universally valid. To more accurately reflect the study's scope and avoid potential misrepresentation, the title should explicitly specify "in vitro" growth kinetics. This clarification would better align the title with the study's actual focus and findings.

Thank you for these suggestions. We have updated the title to include "in vitro" to avoid confusion. The new title now reads, “The capsule and genetic background, rather than specific loci, strongly influence in vitro pneumococcal growth kinetics.” While our study used in vitro data, our goal is to highlight that such in vitro differences in pneumococcal growth may influence in vivo dynamics, as highlighted in several papers referenced in the introduction and discussion.

This study suggests that the entire genetic background significantly influences bacterial growth kinetics. However, to transform these predictions into established facts, extensive experimental validation is necessary. This would involve "bench experiments" focusing on generating and studying mutant variants of serotypes or strains with diverse genomic variations, such as targeted deletions. The growth phenotypes of these mutants should be analyzed, complemented by complementation assays to confirm the specific roles of the deleted regions. These efforts would provide critical empirical evidence to support the findings from the GWAS approach and enhance understanding of the genetic basis of bacterial growth kinetics.

We fully agree with this assessment. As reviewer #1 similarly highlighted, additional genetic manipulation studies would provide further helpful information to assess the impact of altering growth rates on virulence and infection. However, the experimental studies were beyond the scope of this study due to several factors beyond our control. However, we intend to conduct follow-up experimental work to provide additional insights into how the combination of serotypes and genetic background influences pneumococcal growth in vitro and virulence in vivo. Because our current study demonstrates that genetic determinants of pneumococcal growth features are not simply confined to single loci, such experimental validation would require novel wet-lab approaches that consider epistatic interactions.

In the discussion section, the authors state that "the influence of serotype appeared to be higher than the genetic background for the average growth rate" (lines 296-298). Alongside references 13-15, this emphasizes the important role of capsular variability, which is a key determinant of serotypes, in influencing growth kinetics. However, this raises the question: why isn't a specific locus like cps, which is central to capsule biogenesis, considered a strong influencer of growth kinetics in this study?

Thank you for highlighting the point above. Indeed, the capsule biosynthesis (cps) locus is associated with pneumococcal growth kinetics, as seen in the analysis of individual serotypes. However, the cps locus does not come up as a hit in the GWAS because we controlled for the population structure of the pneumococcal strains. The absence of the hits in the cps locus is because serotypes, hence cps loci, tend to be tightly associated with lineages despite occasional capsule switches, which introduce serotypes to different lineages. Therefore, controlling for population structure, which is critical for GWAS analyses, virtually eliminates the detection of potential hits within the cps locus. However, detecting such hits with larger datasets may still be possible. For this reason, we performed a separate analysis of the individual serotypes and lineages shown in Figure 3.

One plausible explanation could be the absence of "elevated signals" for cps in the GWAS analysis. GWAS relies on identifying loci with statistically significant associations to phenotypes. The lack of such signals for cps may indicate that its contribution, while biologically important, does not stand out genome-wide. This might be due to the polygenic nature of growth kinetics, where the overall genetic background exerts a cumulative effect, potentially diluting the apparent influence of individual loci like cps in statistical analyses.

We fully agree with this point. We mentioned in the abstract and discussion that the absence of the signals for specific individual loci within the pneumococcal genome may imply that the growth kinetics are polygenic. We have edited the discussion to emphasise the suggested point.

**Reviewer #3 (Public review):**
This study provides insights into the growth kinetics of a diverse collection of Streptococcus pneumoniae, identifying capsule and lineage differences. It was not able to identify any specific loci from the genome-wide association studies (GWAS) that were associated with the growth features. It does provide a useful study linking phenotypic data with large-scale genomic population data. The methods for the large part were appropriately written in sufficient detail, and data analysis was performed with rigour. The interpretation of the results was supported by the data, although some additional explanation of the significance of e.g. ancestral state reconstruction would be useful. Efforts were made to make the underlying data fully accessible to the readers although some of the supplementary material could be formatted and explained a bit better.

Thank you for the excellent summary of the manuscript. We have added some text to clarify the significance of some approaches, including ancestral state reconstruction and supplementary material.

**Reviewer #1 (Recommendations for the authors):**
(1) Since the PCBN was collected pre and post-vaccine introduction, did the authors stratify their analyses other than Figure 7 (disease correlations) to assess how vaccine status may influence growth rates? Is the assertion in Lines 238-239 supported by the in vitro data?

We have done this analysis. Overall, there was no association between vaccine introduction and pneumococcal growth rates. In lines 238-239, we assumed that in vaccinated populations, the host may be more capable of suppressing bacterial replication due to vaccination. However, there was no in vitro data to back this statement. Therefore, we have edited the statement to remove the text regarding vaccination policy.

We considered vaccination status when analysing the data presented in Figure 7. As mentioned in the legend, we only analysed the dataset collected before vaccine introduction to avoid confounding due to vaccination status. To fully assess the impact of vaccination, we would need additional information besides the date of isolation, including vaccine doses and time since vaccination, which was not available for our study.

(2) Similarly, do any of the growth rate metrics correlate with other aspects of the clinical dataset, like the year of isolation or the sex/age of the patient?

We did not include these assessments in the manuscript, as these aspects of the clinical dataset are mostly related to the patient and not necessarily the intrinsic characteristics of the pneumococcus. However, upon revising the manuscript, we compared the growth characteristics against the vaccination period, and we did not find any statistically significant association. The relationship between pneumococcal growth features of the isolates used in the current study and their corresponding clinical manifestations of invasive disease was described in Arends et al. (10.1128/spectrum.00050-22).

(3) When evaluating the impact of serotype on growth rates, did the directionality of some of the described impacts match with those previously reported in other studies?

We were unable to assess the directionality of the serotype’s impact on growth rates. In part, we did not conduct this analysis because our study used different strains from those used in other studies. Such differences in the genetic backgrounds, growth media, and analytical approaches made assessing the consistencies between the studies difficult.

(4) Did the authors expect that a specific growth metric would be more likely to correlate with specific genetic variants? The reader would benefit from a brief discussion of how the metrics (e.g., maximum growth or lag phase duration) are biologically meaningful beyond the overall growth rate.

We indeed expected that specific growth metrics might correlate with certain genetic variants based on their distinct biological roles. The lag phase duration can potentially reflect the ability of the pneumococcus to adapt to environmental conditions, such as nutrient availability or stress, and may be more influenced by regulatory genes involved in sensing and responding to environmental cues (PMID: 30642990, PMID: 22139505). In contrast, maximum growth rate is more likely to be impacted by core metabolic or biosynthetic genes that control the rate of cell division under optimal conditions (PMID: 31053828). Maximum optical density, which reflects the final cell density, might be shaped by factors related to nutrient utilization efficiency, waste tolerance, or quorum sensing. The duration of the stationary phase is related to the switch from lipoteichoic acids to wall teichoic acids, permitting the initiation of the lytic growth phase (PMID: 239401). It is unclear whether this switch is mediated by external triggers or also by intrinsic features of the pneumococcus. Including this type of analysis allows for a more nuanced understanding of how genetic variants contribute to different physiological aspects of microbial growth. The relevance of the lag phase and the stationary phase in relation to the clinical phenotypes of invasive disease (such as pleural empyema and meningitis) of our pneumococcal isolates has been studied and discussed in Arends et al. (PMID: 35678554). The observed associations are summarized in Table 2 of that article. We have added some text in the discussion on the biological relevance of each bacterial growth metric.

(5) For the GWAS analyses, have similar analyses been performed for other S. pneumoniae collections? Are there known "control" loci that the authors could replicate in the current collection to verify the robustness of the approach?

Others have undertaken GWAS analyses of other S. pneumoniae collections elsewhere. Unlike our study, none of the GWAS analyses elsewhere focused on bacterial growth kinetics. Therefore, considering this is the first GWAS study in pneumococcus and bacteria, in general, to focus on growth kinetics, we do not have “control” loci that we could replicate to verify the robustness of the approach. However, we hope that future studies will be able to utilise our findings to compare their approach as more and more similar analyses of in vitro growth data become available.

(6) Is there a statistical method that could predict the sample size necessary to detect the proposed combinatorial or small contributions from various genetic loci to growth rate? This reviewer is not an expert in statistical genetics but would appreciate an indication of the scale required by future studies to identify these regions.

We are unaware of a statistical approach that could predict sample sizes to detect small or combinatorial effect sizes. However, we intend to conduct simulations in future studies to gain insights into the required sample sizes.

(7) WGS and genome assembly metrics should be provided for each sequenced genome especially since only short-read assemblies were performed. If not already deposited, the assemblies should be deposited for data sharing as well.

We have deposited the sequence reads to the European Nucleotide Archive (ENA) and provided the accession numbers, WGS, and assembly metrics in Supplementary Data 1. We have described the tools used to generate the assemblies from the reads.

(8) Please include the specific ethics approval numbers for the sample collection protocol.

Study procedures were approved by the Medical Ethical committees of the participating hospitals, including a waiver for individual informed consent (file number 2020–6644 Radboudumc).

**Reviewer #3 (Recommendations for the authors):**
Certain aspects of the manuscript could be clarified and extended to improve the manuscript.(1) Introductiona) The authors assume knowledge by the reader on Streptococcus pneumoniae, specifically the genetic diversity of lineages and capsules. This diversity is highlighted in the discussion L368 that there are >100 serotypes. The authors should consider backgrounding the number of serotypes and the importance of serotype switching in these bacteria, as well as explaining the diversity of the lineages (GPSC) that are increasingly used as standard nomenclature for Streptococcus pneumonia.

Thank you for bringing this to our attention. We have included a brief description of the GPSC lineages and capsule switching in the introduction.

b) The last paragraph of the introduction is lengthy and gets into the methods and results of the manuscript. These could be edited down.

We have revised the paragraph to remove the methods and results.

(2) Methodsa) The authors should provide details on the QC undertaken and any exclusion criteria of genomes based on the QC. The supplement material has tabs e.g. read and assembly metrics but unclear how determined and impacted the study.

We utilised all the genomes available for this study, which had in vitro phenotypic data available. We excluded no genomes due to poor sequence quality.

Additional information about the genomes is available from previous studies, which are referenced in the methods section.

b) Why did the authors map draft assemblies to the reference genome for the SNP alignment (from which the ML tree was inferred)? Draft genome assemblies usually contain errors so there is potential for false positive SNPs. Further, there is a lack of perbase quality information using the draft genome assemblies. Given the short read data are available - why were the reads not used as input for snippy (which is the standard input for snippy)? This may have impacted the results reliant on the SNP calls.

We mapped a combination of reads and draft assemblies to the reference genome to generate the SNP alignment using Snippy (https://github.com/tseemann/snippy). For the pneumococcal isolates, we mapped the reads, while for the included outgroup, we mapped the assembly as we did not have sequence reads available. We have edited the methods section to clarify this.

c) SNP alignment. the authors explain the decision to not undertake recombination detection later in the discussion. Did the authors mask any phage or repeat regions? And how was the outgroup S. oralis included in the analyses e.g what genome was used?

We included the outgroup genome in the alignment generated by SNIPPY, which involved generating aligned consensus sequences for each isolate after mapping the reads to the pneumococcal ATCC 700669 reference genome (GenBank accession: NC_011900), as described in the methods. We have now included the accession number for the S. oralis genome, which was used as an outgroup in our phylogenetic analysis. Phages are not typically common in pneumococcal genomes compared to other species. Similarly, although repeats are present in the pneumococcal genome, the consensus in the field is that these do not particularly bias the pneumococcal phylogeny. Therefore, the consensus in the field has been not to explicitly mask these regions as done for highly clonal bacterial pathogens, such as *Mycobacterium tuberculosis*. Overall, our approach to building the phylogenetic tree is robust compared to alternative methods (PMID:29774245).

d) Should the presence/absence of unitigs that were used as the input for the GWAS be included as a supp dataset?

We have now provided the presence/absence matrix for the unitigs used in the analysis as a supplementary dataset available at GitHub (https://github.com/ChrispinChaguza/SpnGrowthKinetics). We have revised the methods section to include a section on data availability.

e) For the annotation of unitigs, the authors used their bespoke script with features from complete public genomes. Please provide accession/ identifying information of the complete genomes (not only the ATCC 700669) reference in the methods. Also, why did the authors choose not to annotate with annotate_hits_pyseer from pyseer?

We annotated the hits using our bespoke script because we understood our approach better and could control the information generated from the script. Annotating with “annotate_hits_pyseer” from pyseer would produce similar results to both approaches, as they compared the unitigs to annotated reference genomes.

(3) Resultsa) The authors could consider providing an overview of the diversity (e.g. lineages and capsules) in the study and contextualising it in the broader context of Streptococcus pneumoniae population genomics. This would help readers who are less familiar with this pathogen to understand the diversity included in this study.

We included this information in the first paragraph of the results section. Considering that population-level analyses based on this dataset have already been published, we have referenced the corresponding papers to provide additional information to readers.

b) Did the timespan of the study pre and post-PCV7 introduction need to be briefly touched on in the results? For example, did the serotypes and lineages vary over the two collection periods and does this need to be considered in the interpretation of the results at all?

The prevalence of serotypes and lineages varied over time, partly due to the introduction of vaccines and random temporal fluctuations in the distribution of strains. We did not explicitly adjust for time, as this is not likely to influence the intrinsic biology of the strains. However, we adjusted for the population structure of the strains, whose changes would most likely affect the distribution of strains in the population. For other analyses, including that in Figure 7, we considered the vaccination status by restricting the analysis to the isolates collected before vaccine introduction.

c) Figures. Some of the figures had very small text (especially Figure 1) that was difficult to read and Figure 2 and Figure 4 were mentioned once, while several paragraphs of results were used to discuss Figure 3. Is Figure 1 required as a main figure? Could Figure 3 be split? e.g. one with the chord diagram, one with panels b-e, and one with panels jq? Figure 4 - the ancestral state reconstruction analyses could be expanded upon in the results.

We have increased the text in some figures where possible. However, for figures that show more information, smaller text is more suitable.

Figure 1 is essential to the manuscript as it provides a visual overview of the approach used in this study. Without this figure, it may be difficult for some readers, especially those unfamiliar with bacterial genomic analyses, to understand our study approach and how we estimated the pneumococcal growth parameters used for the GWAS.

For Figure 4, we prefer to keep it as it is, to have the information in one place, as splitting it will mean including some of the panels in the supplementary material, considering that we already have seven figures in the manuscript.

We have added additional text to the results regarding the ancestral reconstruction analyses. We included them mainly to demonstrate the correlation between the pneumococcal growth rates and the phylogeny.

(4) Discussiona) Why was 15 hours for culture undertaken and not 24? The authors discuss the impact that this may have had on their results.

The 15-hour incubation period was deliberately chosen, as the growth curves indicate that most isolates had reached the stationary phase by that time. Extending the culture duration would likely not have yielded additional meaningful data. As is well established, Streptococcus pneumoniae undergoes autolysis upon reaching a certain cell density, which could distort growth measurements and complicate interpretation if incubation were prolonged. For clarification, we have changed the sentences related to this topic in the Discussion.

b) Some paragraphs in the discussion were very long e.g. L347-381. The authors could consider breaking long paragraphs down into shorter ones to improve the readability of the manuscript.

We agree with this assessment. We initially wanted to include all the information on the study’s limitations in the same paragraph. However, as suggested, we have now split the highlighted paragraph into two shorter paragraphs.

(5) Supplementary Dataa) Providing information in each tab of each supp data file would be useful. For example - including a table header that explained what was in each sheet rather than relying on the tab names. Formatting for some of the underlying supplementary data could be improved e.g. in supplementary data 2 no explanation is given to interpret the data included in these files.

Thank you for the suggestions. For clarity, we have included a header in each tab of the spreadsheet that describes what is included in each dataset. We have also removed the previous Supplementary Data 2. We realised that the information presented in this spreadsheet was redundant, as it was already available in Supplementary Data 1.